# Hallucination Detection from Structural Reasoning Model

**Jianbo Sun** [1]   **Pengkun Yang** [1]

## Abstract

Hallucinations pose a key challenge for large language models, and chain-of-thought prompting exposes intermediate reasoning but usually treats traces as linear sequences, making cross-step dependencies and unsupported intermediate claims difficult to identify. We propose a *structural reasoning model* to describe interactions among local reasoning steps. To detect hallucinations, we extract a directed acyclic reasoning graph over conditions and intermediate claims, verify each claim against its parent nodes, and aggregate the step signals with a simple mass-flow rule. Under a probabilistic erasure-gate abstraction, we interpret this aggregation as measuring information loss along the reasoning graph. Experiments on GSM8K, MATH, HumanEval, and HotpotQA show that the proposed method is most advantageous on longer, dependency-rich reasoning traces such as math and code generation, while remaining competitive on shorter factual QA; these results provide a structured perspective on chain-of-thought evaluation. All code and data are available at https://github.com/soncheinbok/FlowScore.

## 1. Introduction

Large language models (LLMs) are rapidly becoming a general-purpose interface for accessing and synthesizing information, with growing deployment in search, writing, coding, and decision-support workflows; yet they remain prone to hallucinations—generating content that is not supported by the provided evidence and may be factually wrong (Ji et al., 2023; Huang et al., 2025). For many tasks that require multi-step computation, it is now standard to elicit intermediate reasoning traces via chain-of-thought (CoT) prompting

(Wei et al., 2022). While CoT can improve problem-solving performance, an otherwise coherent rationale can still hallucinate facts or slip in invalid deductions (Turpin et al., 2023). In math word problems, a single early mistake may be reused as a premise downstream, so the final solution can look well-formed even when it ultimately rests on an unsupported intermediate claim (Mukherjee et al., 2025). More broadly, although intermediate traces seem to offer a scalable way to monitor model reasoning, recent evidence suggests that this monitoring opportunity can be fragile, especially under distribution shift and long-horizon reasoning, making detectors that rely on superficial properties of a linear trace brittle (Korbak et al., 2025; Guan et al., 2025; Cheng et al., 2025).

Recent work has begun to treat a model's own generations as a practical signal for hallucination detection. A common sampling-based approach is to draw multiple generations and treat disagreement as a warning sign (Manakul et al., 2023; Farquhar et al., 2024; Abdaljalil et al., 2025). These sampling-based uncertainty signals have since been carried into chain-of-thought evaluation by sampling multiple chains and aggregating agreement not only on the final answer but also on the reasoning process, yielding instance-level risk scores for multi-step solutions (Xu et al., 2025; Wang et al., 2026).

While effective in many cases, these signals are still proxies: if the conditional answer distribution is sharply peaked at some $a^\dagger$ (i.e., $\mathbb{P}(\hat{a} = a^\dagger \mid x) \approx 1$), then $K$ samples will be nearly identical even when $a^\dagger \notin \mathcal{A}^\star(x)$, where $\mathcal{A}^\star(x)$ denotes the set of correct answers, so disagreement-based detectors can fail in a "confidently wrong" regime. Moreover, self-reported confidence is often poorly calibrated, and multi-sample procedures incur extra latency/compute (requiring $K$ completions), making them less suitable under tight latency or single-completion monitoring constraints.

Besides self-consistency and confidence-based methods, a second family relies on LLM-as-a-judge methods to evaluate generated content (Liu et al., 2023; Zheng et al., 2023; Kim et al., 2024). While this enables scalable verification, LLM judgments can be noisy and systematically biased, motivating careful evaluation protocols and aggregation (Dubois et al., 2024; Wataoka et al., 2024; Shi et al., 2025). These issues become most pronounced when the judge is

[1]Department of Statistics and Data Science, Tsinghua University, Beijing, China. Correspondence to: Pengkun Yang <yang-pengkun@tsinghua.edu.cn>.

*Proceedings of the 43rd International Conference on Machine Learning*, Seoul, South Korea. PMLR 306, 2026. Copyright 2026 by the author(s).

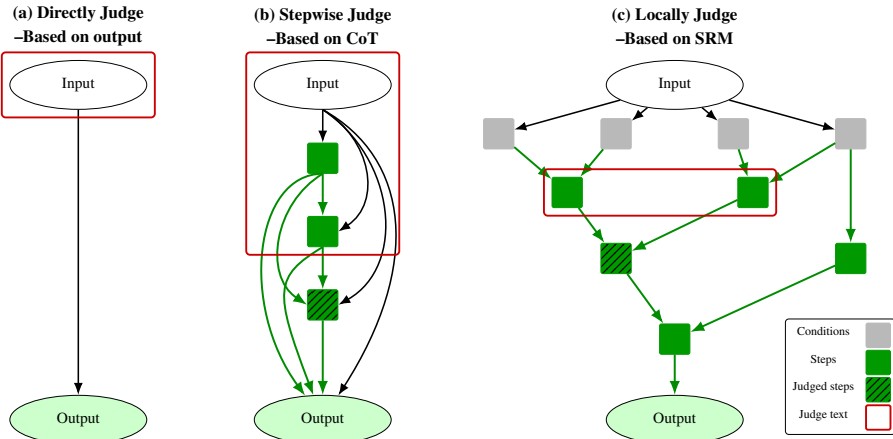

*Figure 1.* Three views of using intermediate signals for hallucination scoring. (a) Judging only the final answer often leaves the verifier without enough supporting evidence. (b) Judging a full linear CoT exposes all prior steps, but when the trace is long, it is hard to pinpoint the relevant support for a step, so the judge can easily miss the key error. (c) Our approach first constructs a structural reasoning model (SRM) as a dependency DAG with explicit source citations, enabling local checks against the cited supports and making hallucinated or unsupported steps easier to detect.

asked to assess the entire reasoning trace in one pass: even long-context LLMs can exhibit position bias and pay less attention to crucial evidence that appears away from the ends of the input (Liu et al., 2024; Yu et al., 2025). As a result, whole-trace judging may overlook a single incorrect step embedded in a long trace, motivating protocols that restrict the judge to minimal, step-specific context and then aggregate local decisions.

A complementary limitation is representational. Standard CoT outputs are long linear traces, where the support for each claim is not explicitly identified and is dispersed across earlier steps; a verifier must infer which earlier statements support each step. This makes faithful evidence attribution and step-level entailment checking challenging in long contexts, even for strong LLM judges. For example, some prefix-based schemes can overweight early steps and dilute errors that appear late in the trace (Liu et al., 2024; Lu et al., 2026). These limitations suggest that effective hallucination detection may require moving beyond linear traces, toward representations that make step-to-support relations explicit.

To make hallucination scoring sensitive to evidence sources and step dependencies, we move from linear reasoning traces to a dependency view of solutions, which we call a *structural reasoning model* (SRM, Figure 1). Using structured prompting, we elicit a reasoning dependency graph where edges record support citations, making dependencies explicit. We then use *local* verification to check each claim against its directly cited supports, producing stepwise signals. Finally, we propagate and aggregate the local step signals on the dependency DAG to produce a flow score, which summarizes how local unreliability accumulates through the graph structure (Wainwright & Jordan, 2008).

This paper is also related to a growing shift toward structured intermediate representations for LLM reasoning and evaluation. Beyond linear CoT, structured prompting frameworks explore multiple reasoning branches (e.g., Tree-of-Thoughts and Graph-of-Thoughts), yielding tree/graph-shaped traces (Yao et al., 2023; Besta et al., 2024). Closer to our setting, PARC rewrites a reasoning chain into a premise-augmented dependency graph by identifying the premises needed for each step (Mukherjee et al., 2025), while Graph of Verification (GoV) verifies graph-ordered reasoning blocks against predecessor evidence (Fang et al., 2026). These approaches clarify premise structure, but they require additional graph rewriting or block construction and can remain sensitive to noisy local checks. Recent work also represents a single solution as a directed acyclic graph (DAG) for reasoning or verification, with nodes as intermediate statements and edges encoding their dependencies (Zhang et al., 2024; 2026); this graph view enables combining local checks without counting the same upstream statement multiple times.

Our contributions are summarized as follows:

(1) **Structural reasoning model (SRM).** We introduce an SRM viewpoint for multi-step solutions by formalizing a stepwise local Markov property and operationalizing it with *structured prompting* that produces an explicit dependency DAG over conditions and intermediate steps. This representation then supports local hallucination detection.

(2) **Flow score for DAG aggregation.** We propose a simple *DAG-based aggregation statistic* that combines noisy, evidence-only step verification signals over the dependency graph, producing an instance-level hallucination score while accounting for shared supports across downstream steps.

(3) **Interpretation and empirical validation.** We provide

an interpretable information-theoretic semantics for the flow score under an idealized erasure-gate abstraction, and validate the framework on GSM8K, MATH, HumanEval, and HotpotQA with sampling-based, judge-based, and graph-based baselines where matched comparisons are available. Controlled studies further analyze judge-matched controls, local verifier choice including Qwen-3.5-9B, DAG quality, reasoning length, structured prompting, aggregation rules, and practical token overhead.

The rest of the paper formalizes SRM and FlowScore (Sections 2–4) and then gives the experimental setup and results (Sections 5–6). The appendix contains proofs, prompts, implementation and cost details, and ablations on evidence locality, verifier strength, DAG quality, prompting, length, and node sensitivity.

## 2. Preliminary

**Step-by-step outputs as an autoregressive model.** We model a black-box LLM as an autoregressive generator. A prompt is sampled from a distribution $\mathcal{P}$, and the model produces a token sequence $x = (x_1, \ldots, x_L) \in \mathcal{V}^L$ by sampling

$$x_i \sim p_\theta(\cdot \mid \text{prompt}, x_{<i}), \qquad i = 1, \ldots, L,$$

where $\mathcal{V}$ is the vocabulary and $\theta$ denotes the model parameters. Since we access the LLM only as a black-box sampler, we suppress the dependence on $\theta$ and write $p$ in place of $p_\theta$ throughout.

By the chain rule,

$$p(x \mid \text{prompt}) = \prod_{i=1}^{L} p(x_i \mid \text{prompt}, x_{<i}).$$

To work at the level of chain-of-thought steps, we treat the response as a sequence of $T$ steps rather than individual tokens. We split the token sequence $x = (x_1, \ldots, x_L)$ into $T$ contiguous blocks. Let $1 = b_1 < b_2 < \cdots < b_{T+1} = L + 1$ be the block boundaries, and define

$$s_t := (x_{b_t}, x_{b_t+1}, \ldots, x_{b_{t+1}-1}), \qquad t = 1, \ldots, T.$$

We denote the resulting step sequence by $s = (s_1, \ldots, s_T)$.

This step representation inherits a conditional distribution from the underlying token-level model. We write this induced step-level distribution as $p(s \mid \text{prompt})$, and model it in the usual autoregressive form

$$p(s \mid \text{prompt}) = \prod_{t=1}^{T} p(s_t \mid \text{prompt}, s_{<t}),$$

where $p(s_t \mid \text{prompt}, s_{<t})$ is the distribution of the next step given the prompt and all previous steps. In our experiments,

step boundaries are enforced by structured prompting, so the step sequence $s$ is directly observable from the model output.

**Structural reasoning model.** In real-world reasoning, producing a new step typically relies on only a small subset of the prior statements, rather than on the entire history. We formalize this intuition in probabilistic terms by positing a conditional-independence structure among steps. For each step index $t \in \{1, \ldots, T\}$, we assume the following local Markov property (conditioning on the observed context prompt):

$$p(s_t \mid \text{prompt}, s_{<t}) = p(s_t \mid \text{prompt}, s_{\text{pa}(t)}), \quad (1)$$

where $\text{pa}(t) \subseteq \{1, \ldots, t-1\}$ denotes one minimal index set for which (1) holds.

Consequently, the step-level distribution admits the factorization

$$p(s \mid \text{prompt}) = \prod_{t=1}^{T} p(s_t \mid \text{prompt}, s_{\text{pa}(t)}).$$

Following Pearl's structural causal model (SCM) formalism (Pearl, 2009), we can represent this structure by a directed acyclic graph (DAG) whose nodes correspond to the step variables $\{s_t\}_{t=1}^T$, and with an edge $s_j \rightarrow s_t$ whenever $j \in \text{pa}(t)$. A Markovian SCM realization introduces mutually independent exogenous noises $\{u_t\}_{t=1}^T$ and structural equations

$$s_t := f_t(\text{prompt}, s_{\text{pa}(t)}, u_t), \qquad t = 1, \ldots, T,$$

which induces an observational distribution that satisfies the above DAG Markov property and factorization.

We call our abstraction a structural reasoning model rather than a structural causal model. Our focus is on a dependency structure elicited by structured prompting. This structure is directly observable from the model output, approximates the target conditional-independence pattern among steps, and supports step-level verification and aggregation. Accordingly, the resulting DAG is used as a reasoning/dependency representation for hallucination risk, rather than as a causal graph to be learned from data via causal discovery or used for estimating interventional effects (Koller & Friedman, 2009).

**Hallucination detection as a hypothesis testing problem.** For each instance, we observe a question $x \in \mathcal{X}$ and a model-generated structured solution $r \in \mathcal{R}$. Let $\hat{a}(r)$ be the final answer extracted from $r$, and let $\mathcal{A}^\star(x)$ denote the set of correct answers. We define the answer-level hallucination indicator

$$y \triangleq \mathbf{1}\{\hat{a}(r) \notin \mathcal{A}^\star(x)\}.$$

**Structured prompt.** The model must output *Conditions*, then *Steps* with explicit *Source* citations.

**Example.**

**Question:** If a store sold 12 apples and then sold 8 more, how many apples were sold in total?

```
Conditions:
C1:  The store sold 12 apples.
C2:  The store sold 8 more apples.
Steps:
S1 (Source:  C1, C2):  Total sold = 12+8 =
20.
Answer (Source:  S1):  20.
```

*Figure 2.* Structured prompting for DAG construction. The model is instructed to output atomic `Conditions` and `Steps`, and to attach a `Source` field to each step that cites its supporting subset of prior statements. We parse these citations to form directed edges from the cited statements to the current step, yielding an acyclic dependency graph. The full prompting template and formatting rules are provided in Appendix B.2.

We consider the binary testing problem

$$H_0 : y = 0 \qquad \text{versus} \qquad H_1 : y = 1.$$

A detector is a test $\phi : \mathcal{X} \times \mathcal{R} \to \{0, 1\}$, where $\phi(x, r) = 1$ denotes rejection of $H_0$. We focus on tests induced by a real-valued test statistic $T : \mathcal{X} \times \mathcal{R} \to \mathbb{R}$, yielding the one-parameter family of critical regions

$$\phi_\tau(x, r) = \mathbf{1}\{T(x, r) \geq \tau\}, \qquad \tau \in \mathbb{R}.$$

For a target level $\alpha \in (0, 1)$, the threshold $\tau$ may be chosen to control the Type I error $\mathbb{P}(\phi_\tau = 1 \mid H_0) \leq \alpha$. In our empirical evaluation, we vary $\tau$ to obtain the operating characteristics of the induced tests; under strong class imbalance, we summarize performance primarily via precision–recall style criteria (equivalently, positive predictive value versus true positive rate) as described in Section 5.2.

## 3. Methodology

**From structured outputs to a dependency DAG.** We elicit a solution with a fixed structured prompt (Fig. 2), where the model outputs (i) *Conditions* $\mathcal{C} = \{c_i\}_{i=1}^m$, (ii) *Steps* $\mathcal{S} = \{s_t\}_{t=1}^T$, and (iii) for each step $s_t$, a *Source* field that cites a supporting subset. We convert this output into a dependency DAG $G = (V, E)$ by taking $V = \mathcal{C} \cup \mathcal{S}$ and adding an edge $(u \to s_t) \in E$ for every cited parent $u \in \mathrm{pa}(s_t)$. Here $\mathrm{pa}(s_t)$ denotes the set of statements explicitly cited in the `Source` field of step $s_t$; we view this set as an observable proxy for the latent minimal parent index set $\mathrm{pa}(t)$ in (1). By construction, citations only point to earlier statements, so $G$ is acyclic.

**Step reliability and practical proxies.** For each step $s_t$, let $q_t \in [0, 1]$ denote its probability of being locally supported conditional on the prompt and its cited parents:

$$q_t \triangleq \mathbb{P}(s_t \text{ is correct} \mid \text{prompt}, \ \mathrm{pa}(s_t)) \in [0, 1].$$

**Algorithm 1** Flow score on an SRM dependency DAG

---

**Require:** DAG $G = (V, E)$; condition nodes $C$; step nodes $S$; reliabilities $q_c = 1$ for $c \in C$ and $q_v \in [0, 1]$ for $v \in S$; source mass $\mu$; routing kernels $P_v$ on $\Gamma^+(v)$.
**Ensure:** $\mathrm{Flow}_{\mu,P}(G, \mathbf{q})$ and $\mathrm{FlowScore}_{\mu,P} = 1 - \mathrm{Flow}_{\mu,P}$.
1: $F_{\mathrm{in}}(v) \leftarrow 0$ for all $v \in V$; $F_{\mathrm{in}}(c) \leftarrow \mu(c)$ for all $c \in C$; $A \leftarrow 0$.
2: Compute a topological order $\pi$ of $G$.
3: **for** $v$ in $\pi$ **do**
4: $\quad A \leftarrow A + (1 - q_v)F_{\mathrm{in}}(v)$; $F_{\mathrm{out}} \leftarrow q_v F_{\mathrm{in}}(v)$.
5: $\quad$ **for** $u \in \Gamma^+(v)$ **do**
6: $\quad\quad F_{\mathrm{in}}(u) \leftarrow F_{\mathrm{in}}(u) + F_{\mathrm{out}}P_v(u)$.
7: $\quad$ **end for**
8: **end for**
9: **return** $\mathrm{Flow} = 1 - A$ and $\mathrm{FlowScore} = A$.

---

Our aggregation is defined over the dependency DAG $G$ via the associated reliability values $\mathbf{q} = (q_1, \ldots, q_T)$ on the step nodes.

In empirical instantiations, $q_t$ is not observed. Instead, we query a step-level verifier on $(\text{prompt}, s_t, \mathrm{pa}(s_t))$ and obtain a step-reliability score $\widehat{q}_t \in [0, 1]$. We then feed $\hat{\mathbf{q}} = (\hat{q}_1, \ldots, \hat{q}_T)$ into the scoring procedure. Implementation details of the verifier protocol are provided in Section 5.3.

**From a dependency DAG to an instance-level test statistic.** Each instance $(x, r)$ induces a dependency DAG $G = (V, E)$ together with a graph-indexed reliability field $\mathbf{q} = \{q_v \in [0, 1]\}_{v \in V}$. For fixed $(G, \mu, P)$, where $\mu$ is a distribution over condition nodes $C$ and $P = \{P_v\}$ are routing distributions over out-neighbors $\Gamma^+(v)$, we define

$$\mathrm{FlowScore}_{\mu,P}(G, \mathbf{q}) \triangleq 1 - \mathrm{Flow}_{\mu,P}(G, \mathbf{q}),$$

where $\mathrm{Flow}_{\mu,P}(G, \mathbf{q})$ is the total surviving mass after propagating unit mass from $C$ and letting each node $v$ absorb a fraction $(1 - q_v)$ of its incoming mass; see Algorithm 1 for the computation of $\mathrm{Flow}_{\mu,P}(G, \mathbf{q})$.

Then, we set

$$T(x, r) \equiv \mathrm{FlowScore}_{\mu,P}(G, \mathbf{q}).$$

Section 4 provides an interpretable justification for $\mathrm{Flow}_{\mu,P}(G, \mathbf{q})$ under idealized assumptions on $\mathbf{q}$, which motivates using its complement $\mathrm{FlowScore}_{\mu,P}(G, \mathbf{q})$ as the test statistic. In our implementation, $\mathbf{q}$ is replaced by a plug-in estimate from step-level verifier outputs; when no edge weights are specified, we use uniform source mass on $C$ and uniform routing over each node's outgoing children, we take $\mu$ uniform on $C$ and $P_v$ uniform on $\Gamma^+(v)$ (cf. Fig. 3).

## 4. Information-Theoretic View of Flow Score

This section provides a minimal semantics for the flow score from Section 3. Under an idealized random-chain erasure

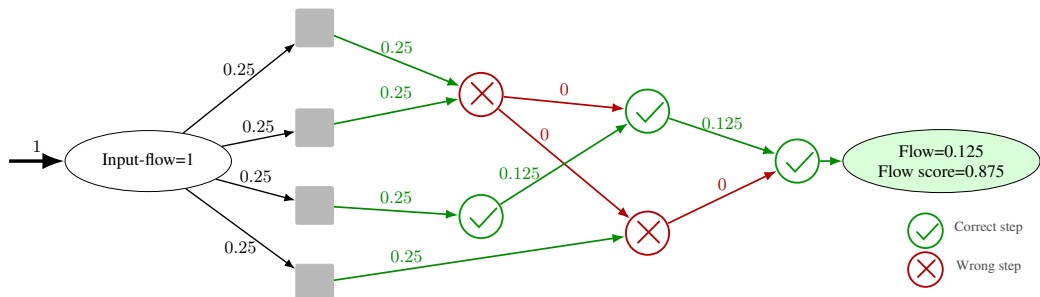

*Figure 3.* Example computation of the DAG-based flow score. Nodes are atomic conditions and intermediate steps; directed edges follow explicit source citations. Step-level verification signals (green/red) are attached to nodes and aggregated via a forward flow: starting from unit input flow, mass is propagated and split across outgoing edges to downstream dependent steps; incorrect steps absorb incoming mass, and the total absorbed mass yields an instance-level hallucination score. In this example, the absorbed mass is 0.875.

abstraction, the flow score equals the probability that the task-relevant content is erased somewhere along the sampled support chain, and equals the fraction of mutual information lost before reaching the terminal sink. This provides an interpretable explanation of the score.

**Theoretical setup.** Fix a reasoning DAG $G = (V, E)$ with a partition $V = C \cup S$, where $C$ are condition nodes (always reliable) and $S$ are step nodes. Each node $v$ has a reliability $q_v \in [0, 1]$, with $q_c = 1$ for all $c \in C$. Let $\mu$ be a distribution on $C$. For a node $v \in V$, let $\mathrm{pa}(v) := \{u : (u \to v) \in E\}$ and $\Gamma^+(v) := \{u : (v \to u) \in E\}$ denote its in- and out-neighborhood in the DAG. Fix an initialization distribution $\mu$ on the condition nodes $C$ and a collection of routing distributions $\{P_v\}_{v \in V}$, where each $P_v(\cdot)$ is supported on $\Gamma^+(v)$ (and $v$ is absorbing if $\Gamma^+(v) = \emptyset$), which defines a Markov kernel on $V$ by $\mathbb{P}(V_{i+1} = u \mid V_i = v) = P_v(u)$. These choices induce a distribution over directed support chains $\Pi = (V_0, \ldots, V_L)$ by sampling $V_0 \sim \mu$ and iterating $V_{i+1} \sim P_{V_i}(\cdot)$ until the first time a sink is reached. Here $\{P_v\}$ is an auxiliary routing rule for sampling an active dependency-respecting chain (restricted to outgoing edges), rather than a conditional distribution of node contents given $\mathrm{pa}(v)$.

**Mutual information** Throughout, $I(A; B)$ denotes the mutual information between random variables $A$ and $B$, defined as

$$I(A; B) = \mathbb{E}_{p(a,b)}\left[\log \frac{p(a, b)}{p(a)p(b)}\right],$$

and $I(A; B \mid C)$ denotes conditional mutual information. In particular, $I(U; X_{V_L}, \Pi)$ is the mutual information between $U$ and the pair $(X_{V_L}, \Pi)$. It measures how much knowing one variable reduces uncertainty about another.

To give the flow score a minimal, computable semantics, we assume that each step node transmits the task-relevant

content with probability $q_v$, and with probability $1 - q_v$ it outputs a fresh noise. This abstraction is consistent with the intuition behind sampling-based detectors (see Section 5.2): supported claims correspond to a concentrated conditional distribution and are therefore stable across samples, whereas unsupported claims correspond to an underdetermined continuation and exhibit higher-entropy variability (e.g., (Manakul et al., 2023)). Formally, under Assumption 4.1, if a step node $v$ is erased then its realized content becomes an i.i.d. noise variable $Z_v$ across independent runs, so the probability of cross-run agreement at that step reduces to the collision probability $\Pr[Z_v = Z'_v] = \sum_z p(z)^2$, which is small for diffuse/high-entropy noise; in contrast, when the gate transmits, the step preserves the upstream content, yielding stability.

**Assumption 4.1** (Random-chain erasure gates). For each step node $v \in S$, there is an independent gate $G_v \sim \mathrm{Bernoulli}(q_v)$, independent of $\Pi$ and of the initial task-relevant content $X_{V_0}$. Let $\{Z_v\}_{v \in S}$ be independent noise variables, jointly independent of $(X_{V_0}, \Pi, \{G_v\})$.

Thus each step node implements an erasure-gate structural equation (an erasure channel) with erasure probability $1 - q_v$, and the randomness in $\Pi$ corresponds to sampling which dependency-respecting support path is "activated".

Define the propagated content $\{X_{V_i}\}_{i=0}^L$ recursively: for $i = 0, \ldots, L - 1$,

$$X_{V_{i+1}} = \begin{cases} (\mathsf{ok}, X_{V_i}), & \text{if } V_{i+1} \in S \text{ and } G_{V_{i+1}} = 1, \\ (\mathsf{er}, Z_{V_{i+1}}), & \text{if } V_{i+1} \in S \text{ and } G_{V_{i+1}} = 0. \end{cases}$$

Conditioned on a realized support chain $\Pi = (V_0, \ldots, V_L)$, the propagated contents $\{X_{V_i}\}_{i=0}^L$ form a simple structural causal model on the chain: $X_{V_0}$ is exogenous (task-relevant content at the start), and for each $i \geq 1$,

$$X_{V_i} := f_{V_i}(X_{V_{i-1}}, G_{V_i}, Z_{V_i}),$$

where

$$f_v(x, g, z) = \begin{cases} (\text{ok}, x), & \text{if } v \in S \text{ and } g = 1, \\ (\text{er}, z), & \text{if } v \in S \text{ and } g = 0. \end{cases}$$

**Theorem 4.2** (Forward flow as survival probability and exact mutual information contraction coefficient)**.** *Under the random-chain erasure abstraction above, define the erasure event*

$$\mathcal{E}_\Pi := \{\exists\, i \geq 1 : V_i \in S,\ G_{V_i} = 0\}.$$

*Then:*

*(i) Forward flow equals survival probability (equivalently, one minus erasure probability).*

$$\text{Flow}_{\mu,P}(G, q) = \mathbb{E}\Big[\prod_{v \in \Pi \cap S} q_v\Big] = \mathbb{P}[\mathcal{E}_\Pi^c], \qquad (2)$$

$$\text{FlowScore}_{\mu,P}(G, q) := 1 - \text{Flow}_{\mu,P}(G, q) = \mathbb{P}[\mathcal{E}_\Pi]. \qquad (3)$$

*(ii) Mutual-information contraction coefficient. For any auxiliary random variable $U$ satisfying $U \to X_{V_0}$,*

$$I(U;\, X_{V_L}, \Pi) \;=\; \text{Flow}_{\mu,P}(G, q)\, I(U;\, X_{V_0}). \qquad (4)$$

*Equivalently, $\text{FlowScore}_{\mu,P}(G, q)$ is the exact fraction of task-relevant mutual information erased before the chain reaches the terminal.*

Polyanskiy and Wu (Polyanskiy & Wu, 2017) develop strong data-processing inequalities (SDPI) for channels and Bayesian networks, and interpret several network contraction bounds via comparisons to erasure/percolation processes. Theorem 4.2 can be viewed as establishing an expected (random-$\Pi$) contraction identity for the end-to-end randomized channel $X_{V_0} \mapsto (X_{V_L}, \Pi)$ under our random-chain erasure abstraction: eliminating $\text{Flow}_{\mu,P}(G, q)$ between (2) and (4) yields

$$\frac{I(U;\, X_{V_L}, \Pi)}{I(U;\, X_{V_0})} \;=\; \mathbb{E}\Big[\prod_{v \in \Pi \cap S} q_v\Big], \qquad (5)$$

i.e., the mutual information contracts by the survival probability of the task-relevant content along a randomly sampled dependency-respecting support chain.

**Corollary 4.3** (Weighted $\ell_1$-Lipschitz continuity)**.** *Fix $(G, \mu, P)$. Let $f(q) := \text{FlowScore}_{\mu,P}(G, q)$ for $q \in [0, 1]^{|S|}$. Define the* visitation probability

$$\rho_u := \mathbb{P}[u \in \Pi \cap S], \qquad u \in S, \qquad (6)$$

*where $\Pi$ is sampled from $(\mu, P)$ as in Theorem 4.2. Then for any $q, q' \in [0, 1]^{|S|}$,*

$$\big|f(q) - f(q')\big| \;\leq\; \sum_{u \in S} \rho_u \big|q_u - q'_u\big|. \qquad (7)$$

In our pipeline, the verifier outputs discrete step-level labels that serve as proxies for $q_v$. Theorem 4.2 therefore provides a concrete interpretation: $\text{Flow}_{\mu,P}(G, q)$ is the survival probability of task-relevant content along a randomly sampled valid support chain, and equals the exact fraction of mutual information retained under this abstraction; equivalently, $\text{FlowScore}_{\mu,P}(G, q)$ is the probability of erasure and the exact fraction of mutual information lost. Because the chain follows graph dependencies, a step that acts as shared support is visited more frequently under $(\mu, P)$ (higher $\rho_u$), and thus has greater influence on the final score; Corollary 4.3 formalizes this sensitivity via a weighted $\ell_1$ bound, implying that noise on high-$\rho_u$ steps is amplified. Appendix E provides a node-perturbation diagnostic for this prediction by flipping labels at different visitation-probability levels.

## 5. Experimental Setup

### 5.1. Datasets and LLMs

We evaluate on four benchmarks that instantiate different forms of multi-step reasoning: GSM8K (Cobbe et al., 2021) for grade-school arithmetic, MATH (Hendrycks et al., 2021) for competition-level mathematics, HumanEval (Chen et al., 2021) for code-generation reasoning with executable tests, and HotpotQA (Yang et al., 2018) for multi-hop factual question answering. This suite lets us compare the same structural interface across arithmetic computation, symbolic mathematics, program synthesis, and factual composition. For each benchmark, we sample model-generated solutions using three LLMs: `gpt-4o-mini`, `gemini-2.5-flash-lite`, and `grok-4-fast-non-reasoning`. For conciseness, in all tables and figures we abbreviate these base models as GPT, GEM., and GROK, respectively. Our goal is to detect cases where these models produce incorrect final answers.

### 5.2. Baseline Methods

In the self-contained black-box setting, existing hallucination detectors can be broadly categorized into two families. The first family derives uncertainty/consistency signals from multiple samples and flags errors when answers or reasoning traces disagree. The second family relies on an LLM-as-a-judge/verifier to monitor and score model outputs—often at the step level—and then aggregates these local assessments into an example-level risk estimate. We include representative baselines from both paradigms.

**Sampling-based method** We consider several representative sampling-based baselines that detect hallucinations via multi-sample uncertainty and consistency. Specifically, we include SelfCheckGPT (SELF) (Manakul et al., 2023), PDS (Xu et al., 2025), and RACE (Wang et al., 2026), which

aggregates several answer and reasoning consistency signals into a unified score; we also report the SINDEX uncertainty indicator proposed by (Abdaljalil et al., 2025).

**Judge-based and graph-based methods** We include three LLM-as-judge baselines and instantiate all of them with the same GPT-5.2 judge used by our step verifier, so their differences mainly reflect evidence organization and aggregation rather than judge family. *Direct judge* queries GPT-5.2 on the entire generated rationale and converts its verdict into an instance-level error score. PARC rewrites a linear chain into premise-augmented reasoning steps and verifies each step against supporting premises (Mukherjee et al., 2025). Graph of Verification (GoV) verifies graph-ordered node blocks against predecessor evidence (Fang et al., 2026); since the released GoV pipeline is designed for its original graph/block construction setting and does not directly provide a matched construction protocol for our four-benchmark SRM setting, we instantiate its verification stage on our high-fidelity dependency graph for a controlled comparison and report this as a GoV proxy. This high-fidelity graph variant is denoted FLOW+.

**Metrics.** Hallucination detection is a rare-event setting; we therefore evaluate primarily on the precision–recall (PR) plane. For a thresholded score $\hat{y}_\tau \in \{0, 1\}$, let $\mathrm{Prec}(\tau) = \mathrm{TP}/(\mathrm{TP}+\mathrm{FP})$ and $\mathrm{Rec}(\tau) = \mathrm{TP}/(\mathrm{TP}+\mathrm{FN})$ denote precision and recall. We also use the $F_1$ score $F_1(\tau) = \frac{2\,\mathrm{Prec}(\tau)\mathrm{Rec}(\tau)}{\mathrm{Prec}(\tau)+\mathrm{Rec}(\tau)}$. For score-valued methods, we sweep thresholds to obtain PR curves and summarize ranking quality by average precision (AP), i.e., a standard summary of the precision–recall curve. On PR plots we overlay iso-$F_1$ contours and report $\max F_1 := \max_\tau F_1(\tau)$ to illustrate achievable operating tradeoffs. We do not emphasize ROC-AUC under severe imbalance. For continuous-valued methods, PR plots show threshold sweeps. For binary or low-cardinality judge outputs, AP is computed from the discrete score levels for table comparability, and the corresponding PR plots show the attainable operating points rather than interpolating them as continuous curves.

### 5.3. Implementation Details

**Structured rationale sampling and DAG construction.** For each problem, we prompt the base LLM with a fixed template to produce a single structured solution (Fig. 2; full template in Appendix B.2). The output consists of (i) atomic `Conditions`, (ii) a sequence of `Step t` statements, and (iii) a `Source` field for each step that cites its minimal supporting subset of prior statements (e.g., `Condition 1 & Step 2`). We parse the structured fields deterministically and construct a directed graph by adding edges from each cited statement in `Source` to the current step, yielding an acyclic dependency DAG. We use greedy decoding

(temperature 0.0) for this single-sample pipeline.

**Step-level verification.** Given a constructed graph, we verify each step using an evidence-only judging protocol. For a target step, the Evidence consists of the statements referenced in its Source field (conditions and cited prior steps), and we additionally provide the original question as Background to resolve implicit context. We query GPT-5.2 once per step and require it to output a short reason together with a discrete label $e_t$ in $\{0, 0.5, 1\}$, where $e_t = 0$ indicates entailed/correct, $e_t = 0.5$ indicates insufficient information/uncertain, and $e_t = 1$ indicates incorrect/contradicted; we then set $\hat{q}_t = 1 - e_t$. All verifier calls use temperature 0.0. To test dependence on this verifier, Appendix D.2 reruns the same evidence-only protocol with a local Qwen-3.5-9B verifier and a weaker GPT-4o verifier, showing that Qwen remains competitive while weak local verification sharply degrades AP.

**Experimental details, cost, and ablations.** Full prompts, parsing rules, leakage controls, baseline implementations, and compute settings are given in Appendices B.2, B.3, and B.4. Since FLOW makes one verifier call per step, its overhead is dominated by local verification; Appendix B.5 reports average step counts and prompt-token proxies, including a direct whole-solution judge comparison. Appendix D then reports controlled ablations on evidence locality, verifier choice, graph quality, prompting effects, reasoning length, and node influence. Once the graph and step labels are fixed, parsing, flow computation, and metric calculation are deterministic.

## 6. Results and Analysis

### 6.1. Local Verification Improves Performance

Table 1 summarizes average precision (AP; reported in %) across arithmetic, mathematical, code-generation, and factual multi-hop benchmarks. Figure 4 shows representative precision–recall curves; additional PR curves are provided in Appendix H. Recall that FLOW uses the parsed SRM dependency graph, FLOW+ uses the high-fidelity dependency graph, and the GoV proxy applies the Graph-of-Verification protocol on the same high-fidelity graph. On tasks whose error signal is primarily internal to the generated reasoning or code, namely GSM8K, MATH, and HumanEval, FLOW+ achieves the best average AP: 77.70 on GSM8K, 76.59 on MATH, and 83.08 on HumanEval. Its margins over the strongest baseline that is neither FLOW nor FLOW+ are 13.74 AP points on GSM8K, 13.80 on MATH, and 5.26 on HumanEval, indicating that explicit dependency structure and local verification are particularly effective when correctness can be assessed from arithmetic, mathematical, or program-semantic consistency. HumanEval further ex-

poses a limitation of sampling-based consistency signals: the four sampling baselines reach only 14.93–25.52 average AP, whereas FLOW+ reaches 83.08. Manual inspection suggests that many HumanEval failures involve small boundary-condition or semantic changes, such as using `n=1` where `n=0` is required, which can remain nearly consistent at the surface level and therefore be missed by semantic agreement across samples.

HotpotQA shows a different pattern. PDS obtains the highest average AP on HotpotQA (88.69), and SINDEX is slightly above FLOW+ (83.43 vs. 83.23), while RACE is essentially tied with it (83.01). This is consistent with the nature of HotpotQA: the extracted traces are often shorter and more fact-retrieval-oriented, and our evidence-only verifier is not coupled to an external knowledge base or retrieved passages, so local factual judgments can depend strongly on the judge model's internal knowledge. In this regime, sample-level uncertainty can provide a complementary signal about factual instability. Nevertheless, among judge-based and graph-structured methods, FLOW+ remains competitive: it exceeds the GoV proxy on HotpotQA, is slightly above DIRECT, and trails only PARC among judge-based and graph-structured baselines.

The comparison with judge-matched and graph-structured controls helps separate structure from judge strength. DIRECT uses the same GPT-5.2 judge family as the step verifier but scores the whole rationale; it is weaker than FLOW+ on every average row (e.g., 41.55 vs. 77.70 on GSM8K and 46.95 vs. 83.08 on HumanEval). FLOW+ also improves over the GoV proxy in all 12 dataset–model cells, suggesting that aggregation matters under comparable graph-structured verification. Appendix D.1 makes the same-verifier control explicit on Grok/MATH: whole-trace judging stays near 53 AP, prefix-step judging reaches 57.82/62.95 AP, and Step-FLOW reaches 65.70/72.87 AP with base/high-fidelity DAGs. Appendix D.2 shows the complementary limitation: Qwen-3.5-9B remains competitive, whereas weaker GPT-4o sharply degrades under the same protocol. Thus SRM is not tied to GPT-5.2, but reliable local verification is required; Appendix E further shows larger Flow-MaxErr gaps on longer traces.

*Table 1.* Average precision (AP, %) across all benchmarks. SELF, SINDEX, PDS, and RACE are sampling/consistency baselines; DIRECT is a whole-trace GPT-5.2 judge; PARC and GoV are graph-structured judge baselines, with GoV instantiated through the high-fidelity-DAG proxy described in Section 5.2; FLOW and FLOW+ are our raw-DAG and high-fidelity-DAG variants. * denotes binary methods, whose AP values should be interpreted with care; the PR curves and attainable operating points in Fig. 4 and Appendix H provide complementary reference.

| Data | Model | SELF | SINDEX | PDS | RACE | PARC* | GoV* | DIRECT* | FLOW | FLOW+ |
|---|---|---|---|---|---|---|---|---|---|---|
| GSM8K | GPT | 15.13 | 26.16 | 26.94 | 71.48 | 58.40 | 65.00 | 45.33 | 65.76 | **80.27** |
| | Gem. | 9.84 | 22.46 | 28.85 | 57.90 | 63.75 | 56.55 | 39.88 | **74.28** | 73.41 |
| | Grok | 7.53 | 29.82 | 28.45 | 62.50 | 56.94 | 67.87 | 39.44 | 65.42 | **79.43** |
| | Avg. | 10.83 | 26.15 | 28.08 | 63.96 | 59.70 | 63.14 | 41.55 | 68.49 | **77.70** |
| MATH | GPT | 27.32 | 26.81 | 43.63 | 42.47 | 65.40 | 60.17 | 50.62 | 63.36 | **78.30** |
| | Gem. | 25.38 | 46.85 | 56.15 | 61.66 | 64.29 | 50.20 | 25.45 | 72.71 | **76.10** |
| | Grok | 23.35 | 38.66 | 44.54 | 51.92 | 58.67 | 54.86 | 44.15 | 65.70 | **75.38** |
| | Avg. | 25.35 | 37.44 | 48.11 | 52.02 | 62.79 | 55.08 | 40.07 | 67.26 | **76.59** |
| HotpotQA | GPT | 68.30 | 85.00 | **90.16** | 83.83 | 85.18 | 79.29 | 83.40 | 82.72 | 82.37 |
| | Gem. | 69.69 | 86.19 | 89.98 | 85.80 | **90.75** | 80.32 | 82.30 | 84.81 | 85.87 |
| | Grok | 63.59 | 79.10 | **85.92** | 79.41 | 83.99 | 71.57 | 76.70 | 80.09 | 81.44 |
| | Avg. | 67.19 | 83.43 | **88.69** | 83.01 | 86.64 | 77.06 | 80.80 | 82.54 | 83.23 |
| HumanEval | GPT | 14.14 | 13.63 | 19.41 | 16.92 | 39.22 | 79.85 | 57.45 | 77.72 | **84.52** |
| | Gem. | 17.11 | 17.28 | 18.28 | 34.55 | 23.03 | 81.24 | 30.63 | 81.87 | **82.26** |
| | Grok | 13.53 | 19.00 | 16.53 | 25.08 | 33.45 | 72.38 | 52.77 | 69.84 | **82.47** |
| | Avg. | 14.93 | 16.64 | 18.07 | 25.52 | 31.90 | 77.82 | 46.95 | 76.48 | **83.08** |
| Overall Avg. | | 29.58 | 40.92 | 45.74 | 56.13 | 60.26 | 68.28 | 52.34 | 73.69 | **80.15** |

## 6.2. Structural Signal in Score Aggregation

We next separate the effect of the aggregation rule from the effect of the local verifier. In Table 2, each row fixes the generated trace, parsed DAG, and step-level verifier outputs, and changes only how local scores are summarized into an instance-level risk score. FLOW propagates mass forward from the question-side conditions to downstream answer nodes. InvFlow is the reverse diagnostic: it propagates mass backward from answer-side nodes toward the question-side supports; the detailed recursion is given in Appendix C. FLOW is best or near-best in most settings and has the strongest overall average among the listed aggregators, while preserving the forward information-loss semantics from Section 4. MeanErr can be very competitive when many local errors are diffuse, and InvFlow is especially close on HotpotQA where answer-proximal evidence

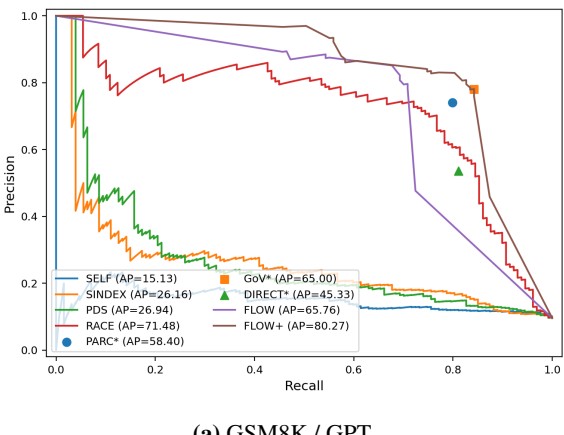 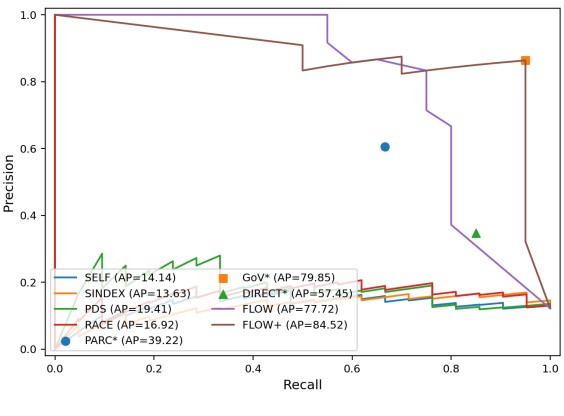

**(a)** GSM8K / GPT        **(b)** HumanEval / GPT

*Figure 4.* Representative precision–recall curves with iso-$F_1$ contours. Continuous-valued methods are shown as thresholded PR curves; binary or ternary judge/graph baselines are shown as attainable operating points. The GSM8K/GPT panel shows the arithmetic setting, while the HumanEval/GPT panel shows a code-generation setting where structural aggregation is especially effective. Additional curves appear in Appendix H.

*Table 2.* Aggregation comparison on the same parsed DAG and verifier outputs (AP, %). Each row fixes generated traces and step labels and changes only the aggregation rule.

| Data | Model | FLOW | InvFlow | MeanErr | MaxErr | MinErr |
|---|---|---|---|---|---|---|
| GSM8K | GPT | **65.76** | 61.49 | 64.55 | 59.75 | 9.63 |
| | Gem. | **74.28** | 56.45 | 62.30 | 48.35 | 14.40 |
| | Grok | 65.42 | 55.65 | **82.43** | 53.45 | 18.07 |
| MATH | GPT | 63.36 | 58.37 | **67.68** | 54.39 | 26.76 |
| | Gem. | 72.71 | 62.68 | **75.23** | 54.28 | 31.86 |
| | Grok | **65.70** | 57.82 | 61.98 | 54.05 | 20.64 |
| HotpotQA | GPT | **82.72** | 82.71 | 81.78 | 76.46 | 64.94 |
| | Gem. | **84.81** | 84.77 | 82.95 | 76.51 | 65.86 |
| | Grok | 80.09 | 80.09 | **80.50** | 72.82 | 61.38 |
| HumanEval | GPT | **77.72** | 61.33 | 77.31 | 43.03 | 29.76 |
| | Gem. | **81.87** | 77.40 | 75.79 | 25.88 | 15.59 |
| | Grok | **69.84** | 60.11 | 66.87 | 45.49 | 21.50 |
| Overall Avg. | | **73.69** | 66.57 | 73.28 | 55.37 | 31.70 |

*Table 3.* AP(%) on MATH. AGN.: structure-agnostic features from $\hat{e}$ only. DAG: features using $(\hat{e}, A)$. F: FULL setting. NS: NOSHORTCUT setting.

| FEAT. | GPT | GEM. | GROK | AVG. |
|---|---|---|---|---|
| AGN. (F) | 75.80 | 79.00 | 69.43 | 74.74 |
| DAG (F) | 77.37 | 80.70 | 74.60 | 77.56 |
| AGN. (NS) | 74.29 | 75.07 | 69.58 | 72.98 |
| DAG (NS) | 77.12 | 77.54 | 73.50 | 76.05 |

from 72.98 to 76.05. Together with Table 2 and the graph-source studies in Appendix D.3, these analyses support a two-part conclusion: the graph controls what evidence each local verifier sees, and the aggregation rule controls how the remaining local noise is propagated to the final risk score.

## 7. Discussion and Future Work

Our findings align with a broader shift toward making model reasoning more inspectable: from eliciting intermediate traces (Wei et al., 2022), to structured exploration over thoughts (Yao et al., 2023), and to scalable evaluation with LLM judges (Kim et al., 2024). The key opportunity is to move hallucination detection beyond instance-level risk scores, toward detectors that localize fragile premises and dependencies. Progress requires both more reliable local signals—through calibration, bias-aware judging, and routing borderline cases to stronger checks—and stronger evidence grounding so that "unsupported" reflects genuine gaps rather than missing implicit context (Mukherjee et al., 2025). A complementary direction is to co-design generation harnesses with verification, so that future systems jointly optimize solution quality, citation fidelity, and verifiability rather than treating structured generation and checking as separate modules.

dominates. MaxErr and MinErr are useful stress tests but are brittle: MaxErr overreacts to isolated local flags, whereas MinErr often ignores downstream failures.

After comparing aggregation rules, we ask whether the dependency matrix itself contains information beyond the unordered list of local verifier labels. Let $\hat{\mathbf{e}} = (\hat{e}_1, \ldots, \hat{e}_S) \in \{0,1\}^S$ denote step-level error indicators, where $\hat{e}_i = 1$ means step $i$ is flagged as incorrect, and let $A$ be the dependency-DAG adjacency matrix induced by Source. We compare structure-agnostic summaries of $\hat{\mathbf{e}}$ with DAG-aware features of $(\hat{\mathbf{e}}, A)$. We fit a regularized logistic regression to predict the instance label $y$ and evaluate by AP on MATH; the feature definitions and evaluation protocol are in Appendix F.1.

Table 3 shows that the adjacency matrix $A$ provides complementary signal beyond structure-free summaries of the same step labels. In the FULL setting, DAG-aware features improve average AP from 74.74 to 77.56; in the stricter NOSHORTCUT setting, the gain remains, improving AP

## Acknowledgements

The authors thank all reviewers for their valuable comments and suggestions. P. Yang is supported in part by National Key R&D Program of China 2024YFA1015800, Tsinghua University Dushi Program.

## Impact Statement

We discuss the potential broader impact of our work, including ethical aspects and possible societal consequences. Our work studies hallucination detection for multi-step reasoning, which may improve reliability and safety when LLMs are used in high-stakes settings. At the same time, misuse or over-reliance on automated detectors could create false confidence; we therefore recommend human oversight and careful calibration when deploying such systems.

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

## A. Proofs

*Proof of Theorem 4.2.* We prove (i) and (ii) separately.

**Proof of (i).** We couple the forward recursion with the random chain $\Pi = (V_0, \ldots, V_L)$. Let $\tau(v) := \inf\{i \geq 0 : V_i = v\}$ (with $\tau(v) = \infty$ if never hit), and define

$$S_i := \prod_{j=1}^{i} G_{V_j} \quad \text{(empty products equal 1)}.$$

We claim that, for every node $v$,

$$F_{\text{in}}(v) = \Pr\big(\tau(v) < \infty, \ S_{\tau(v)-1} = 1\big), \tag{8}$$

i.e., $F_{\text{in}}(v)$ is the probability that the chain reaches $v$ *before any erasure occurs*.

The claim holds for $c \in C$ since $F_{\text{in}}(c) = \mu(c)$. Assume it holds for all parents of $v$. Reaching a parent $u$ with $S_{\tau(u)-1} = 1$ occurs with probability $F_{\text{in}}(u)$, survives through $u$ with probability $q_u$, and transitions to $v$ with probability $P_u(v)$, contributing $q_u F_{\text{in}}(u) P_u(v)$. Summing over parents matches exactly the forward update, proving (8).

For a step node $v \in S$, conditioned on the event $\{\tau(v) < \infty, \ S_{\tau(v)-1} = 1\}$, the *first* erasure occurs at $v$ with probability $(1 - q_v)$, so

$$\Pr(\text{first erasure at } v) = (1 - q_v) \, F_{\text{in}}(v).$$

Summing over $v \in S$ yields

$$\Pr(\mathcal{E}_\Pi) = \sum_{v \in S} (1 - q_v) \, F_{\text{in}}(v) = 1 - \text{Flow}_{\mu, P}(G, q), \tag{9}$$

which is exactly (3). Finally, by independence of the gates,

$$\Pr(\mathcal{E}_\Pi^c) = \mathbb{E}\left[ \prod_{v \in \Pi \cap S} q_v \right],$$

and together with (3) this gives (2).

**Proof of (ii).** Consider a single erasure gate: for any $(U, X)$ and independent $G \sim \text{Bernoulli}(q)$, let $Y = (\text{ok}, X)$ if $G = 1$ and $Y = (\text{er}, Z)$ if $G = 0$, with $Z \perp (U, X)$. Then $I(U; Y) = q \, I(U; X)$ because $Y$ is independent of $U$ on $\{G = 0\}$ and invertible in $X$ given $\{G = 1\}$.

Conditioned on a realized chain $\Pi$, composing these gates along visited step nodes multiplies mutual information by $\prod_{v \in \Pi \cap S} q_v$, hence

$$I(U; X_{V_L} \mid \Pi) = \Big( \prod_{v \in \Pi \cap S} q_v \Big) I(U; X_{V_0}).$$

Since $\Pi$ is sampled independently of $(U, X_{V_0})$,

$$I(U; X_{V_L}, \Pi) = \mathbb{E}[I(U; X_{V_L} \mid \Pi)]$$

$$= \mathbb{E}\left[ \prod_{v \in \Pi \cap S} q_v \right] I(U; X_{V_0}).$$

Substitute (2) to obtain (4). $\qquad\square$

*Proof of Corollary 4.3.* By (3) and (2),

$$f(q) = 1 - \mathbb{E}\left[ \prod_{v \in \Pi \cap S} q_v \right].$$

Hence

$$|f(q) - f(q')| = \left| \mathbb{E}\left[ \prod_{v \in \Pi \cap S} q_v \right] - \mathbb{E}\left[ \prod_{v \in \Pi \cap S} q_v' \right] \right| \leq \mathbb{E}\left[ \left| \prod_{v \in \Pi \cap S} q_v - \prod_{v \in \Pi \cap S} q_v' \right| \right].$$

Fix a realized chain $\Pi$ and write $I := \Pi \cap S$. For numbers $\{a_v\}_{v \in I}, \{b_v\}_{v \in I} \subset [0, 1]$, a telescoping expansion gives

$$\left| \prod_{v \in I} a_v - \prod_{v \in I} b_v \right| \leq \sum_{u \in I} |a_u - b_u|.$$

Applying this with $a_v = q_v$ and $b_v = q'_v$ yields

$$\left| \prod_{v \in \Pi \cap S} q_v - \prod_{v \in \Pi \cap S} q'_v \right| \leq \sum_{u \in S} \mathbf{1}\{u \in \Pi \cap S\} |q_u - q'_u|.$$

Taking expectations and using (6) gives (7). $\qquad\square$

## B. Experimental Details

This appendix first documents the data, labels, prompting protocol, implementation choices, and practical cost. We place these details before the additional results so that the ablations can be read as controlled changes to a fixed pipeline.

### B.1. Datasets and Label Construction

**Benchmarks.** GSM8K contains grade-school arithmetic word problems with natural-language reference solutions and final numeric answers (Cobbe et al., 2021). We evaluate on the standard test split and use it as a setting where errors typically arise from arithmetic or bookkeeping. MATH contains competition-level mathematics problems across algebra, geometry, number theory, and counting/probability (Hendrycks et al., 2021); following prior step-verification work (Xu et al., 2025), we evaluate on a fixed 1,000 problem subset of the official test split. HotpotQA evaluates multi-hop factual question answering (Yang et al., 2018). HumanEval evaluates code-generation correctness with executable unit tests (Chen et al., 2021). In all benchmarks, a positive instance is one whose extracted final answer is incorrect under the dataset-specific checker.

Table 4. Positive rate of error labels. Each entry reports #positives / #total, where positives correspond to golden_label=1.

| Dataset | Gem. | GPT | Grok |
|---|---|---|---|
| GSM8K | 77/1318 | 127/1319 | 77/1319 |
| MATH | 183/996 | 245/999 | 201/1000 |
| HotpotQA | 363/556 | 360/563 | 340/556 |
| HumanEval | 10/161 | 20/164 | 23/164 |

**Answer adjudication.** For GSM8K, we normalize numeric answers by stripping commas, currency symbols, and incidental whitespace, and manually audit residual ambiguous cases. For MATH, where equivalent answers may be written in different algebraic forms, we use an auxiliary equivalence checker (GLM-5) that receives the question, reference answer, and extracted model answer, and returns whether the two answers are equivalent. HotpotQA is evaluated by answer-string normalization and exact/F1-style matching following the dataset convention. HumanEval is evaluated using the official executable unit tests.

### B.2. Prompts and Graph Construction

Our pipeline uses strict, role-specific prompt templates so that parsing is deterministic. The structured-rationale prompt elicits Conditions, numbered Steps, and a Source field for each step (Figure 8). The graph parser adds an edge from each cited source to the current step, producing an acyclic dependency graph. The source-tracing module used by FLOW+ expands and sanitizes parent sets while preserving the same generated solution and verifier. Importantly, this high-fidelity DAG construction never uses the reference answer, executable result, or instance-level correctness label; it only rewrites source citations from the generated structured solution.

**Verifier labels and score convention.** The step verifier returns an error score

$$e_t \in \{0, 0.5, 1\},$$

where $e_t = 0$ means the step is entailed/correct, $e_t = 0.5$ means insufficient information or uncertainty, and $e_t = 1$ means incorrect or contradicted. The flow recursion uses the corresponding reliability proxy

$$\hat{q}_t = 1 - e_t.$$

This convention keeps the theory and implementation aligned: $q_t$ denotes reliability, while $e_t$ denotes local error evidence.

**Leakage control.** For local verification, the judge receives the original question as background, the target step, and only the cited parent statements as evidence. The reference answer, dataset label, and execution/test outcome are never shown to the verifier. We format the terminal answer as a normal graph node and treat its verification identically to other steps. We also log verifier outputs and parsed adjacency matrices for auditability.

### B.3. Baseline Implementation Details

We group baselines by the signal they expose. SelfCheckGPT (SELF) measures cross-sample support using the BERTScore-based variant (Manakul et al., 2023). SINDEX measures answer-set instability by clustering sampled answers (Abdaljalil et al., 2025). PDS/R2PE follows the NLI-based step-verification setup of Xu et al. (2025), instantiated with `roberta-large-mnli`. RACE follows the released scorer pipeline of Wang et al. (2026): we use `sentence-transformers/all-MiniLM-L6-v2` for answer uncertainty, `potsawee/deberta-v3-large-mnli` for reasoning-consistency NLI comparisons, and `bebr2/RACE-CoT-Extractor-Llama-8B` as the CoT extractor. To adapt to lengthy generations, we set the SINDEX clustering similarity threshold to $0.9$. We omit RACE's entity-coherence subscore because entity extraction is less aligned with math/code reasoning and introduces additional noise.

Direct judge, PARC, and GoV are LLM-as-judge baselines with different evidence organizations. Direct judge evaluates the whole generated rationale at once. PARC rewrites a chain into premise-augmented reasoning steps and checks steps against their supporting premises (Mukherjee et al., 2025). GoV verifies graph-ordered node blocks against predecessor evidence (Fang et al., 2026); since the released GoV implementation is tied to its original benchmark and graph/block construction setting, we instantiate its verification stage on the same high-fidelity DAG used by FLOW+ for a controlled comparison and report it as a GoV proxy.

### B.4. Implementation and Reproducibility

**Decoding and parsing.** Unless otherwise stated, the single-sample SRM pipeline uses temperature $0.0$ and provider-default decoding settings beyond temperature. To avoid overstating support from noisy graph annotations, we apply conservative input normalization: only well-formed in-range source references are used, malformed references are ignored, source lists are padded or truncated to match the reasoning trace, and empty sources are treated as unsupported nodes that receive zero incoming reliability mass. Core labels are not imputed; missing or invalid instance/step labels cause the record to be rejected. Once the graph and step labels are fixed, all scalar scores are deterministic.

**Multi-sample baselines.** For sampling-based baselines, we draw $K = 5$ independent traces per problem with temperature $1.0$ and a 2048-token cap. Random seeds are not fully controllable for black-box APIs, so we cache raw generations, model identifiers, timestamps, parsed fields, and all post-processed scores.

**Judge configuration and secondary audit.** The primary step verifier is GPT-5.2 with temperature $0.0$ and a short output budget. Transient API failures are retried up to five times with exponential backoff; after five failed attempts, the judgment is conservatively set to label 1. During the audit, we observed occasional degraded local judgments from the gateway-served GPT-5.2. Therefore, every local verifier output with $e_t = 1$ was rechecked by GEMINI-3-PRO-PREVIEW using the same evidence-only prompt, label space, and leakage controls. When the second verifier assigned a non-error label, we replaced the original $e_t = 1$ with the audited label. This secondary audit never used the reference answer, executable result, or instance-level correctness label. Prompts, cache keys, original labels, audited labels, and error traces are saved for reruns.

**Compute.** API-based judging does not require specialized hardware. Local baseline inference is run on a Linux server with 10 vCPUs, 60GB RAM, and $4\times$ NVIDIA RTX 4090 GPUs. The released artifacts include dependency versions, model identifiers, and scripts for reproducing the deterministic post-processing.

## B.5. Cost and Practical Overhead

The flow recursion itself is a lightweight deterministic graph traversal; the dominant cost comes from generating structured traces and verifying individual steps. Table 5 reports average prompt-token proxies per example, averaged over base models. For reference, we also include a direct whole-solution judge baseline, computed by filling the whole-solution judging prompt with the question and the concatenated model reasoning trace. Under this proxy, the structured FLOW pipeline requires more prompt tokens than a single whole-solution judgment, because it first elicits an explicit support graph and then verifies each step separately. This additional cost is the price of obtaining localized step labels and explicit support diagnostics rather than only an instance-level accept/reject decision.

*Table 5.* Average prompt-token proxy per example from released records, averaged over base models. The direct whole-solution judge uses one prompt containing the question and concatenated model reasoning trace.

| Dataset | Steps | Generation | Step Verifier | Total | Whole Judge |
|---|---|---|---|---|---|
| GSM8K | 4.23 | 190 | 498 | 687 | 218 |
| MATH | 6.15 | 254 | 730 | 983 | 263 |
| HotpotQA | 3.67 | 97 | 241 | 338 | 174 |
| HumanEval | 7.15 | 528 | 1614 | 2143 | 473 |

# C. Inverse-Flow Diagnostic

The main score propagates mass from problem-side conditions toward the terminal answer. As a diagnostic, we also consider reverse propagation: initialize unit mass at the terminal answer node and route mass backward through cited parents. This emphasizes answer-proximal supports rather than premise-to-answer propagation and therefore tests whether the forward direction is merely an arbitrary weighting scheme.

---

**Algorithm 2** Inverse flow score on a dependency DAG

---

**Require:** DAG $G = (V, E)$ with condition nodes $C$, step nodes $S$, terminal answer node $a$, reliabilities $q_c = 1$ for $c \in C$ and $q_v \in [0, 1]$ for $v \in S$, reverse routing kernels $\widetilde{P}_v$ supported on $\Gamma^-(v) = \{u : (u \to v) \in E\}$.
**Ensure:** $\mathrm{InvFlow}(G, q)$ and $\mathrm{InvFlowScore}(G, q) = 1 - \mathrm{InvFlow}(G, q)$.
1: $R_{\mathrm{in}}(v) \leftarrow 0$ for all $v \in V$; $R_{\mathrm{in}}(a) \leftarrow 1$; $A \leftarrow 0$.
2: Compute a reverse topological order $\bar{\pi}$ of $G$.
3: **for** $v$ in $\bar{\pi}$ **do**
4:     $A \leftarrow A + (1 - q_v)R_{\mathrm{in}}(v)$; $R_{\mathrm{out}} \leftarrow q_v R_{\mathrm{in}}(v)$.
5:     **for** $u \in \Gamma^-(v)$ **do**
6:        $R_{\mathrm{in}}(u) \leftarrow R_{\mathrm{in}}(u) + R_{\mathrm{out}} \widetilde{P}_v(u)$.
7:     **end for**
8: **end for**
9: **return** $\mathrm{InvFlow} = 1 - A$ and $\mathrm{InvFlowScore} = A$.

---

*Table 6.* AP(%) comparison between forward flow and inverse flow. Bold indicates the best method within each dataset block.

| Dataset | Method | GPT | Gem. | Grok | Avg. |
|---|---|---|---|---|---|
| GSM8K | Flow | 65.76 | **74.28** | 65.42 | 68.49 |
| | Flow+ | **80.27** | 73.41 | **79.43** | **77.70** |
| | InverseFlow | 61.49 | 56.45 | 55.65 | 57.86 |
| | InverseFlow+ | 69.91 | 62.23 | 69.59 | 67.24 |
| MATH | Flow | 63.36 | 72.71 | 65.70 | 67.26 |
| | Flow+ | **78.30** | **76.10** | **75.38** | **76.59** |
| | InverseFlow | 58.37 | 62.68 | 57.82 | 59.62 |
| | InverseFlow+ | 68.91 | 63.76 | 65.01 | 65.89 |

Reverse propagation is useful as a diagnostic but is weaker than the forward FLOW+ variant on average. This is consistent with the information-loss interpretation in the main text: premise-to-answer propagation better matches how unsupported upstream content contaminates downstream reasoning.

# D. Ablation Studies

This section provides controlled ablations for the SRM pipeline. We study four factors: evidence locality, verifier choice, graph construction, and structured prompting. We also examine reasoning length and node-level sensitivity.

Unless stated otherwise, the generated solution is fixed when an ablation changes the verifier or the aggregation rule. The reference answer and the instance-level correctness label are not given to the step verifier. The verifier receives the question as background, the target step, and the cited parent statements. The verifier then produces a local error label, which is converted to a reliability score and aggregated on the DAG. All AP values are reported in percent. Higher AP is better.

## D.1. Ablation 1: Linear Judge, Removing DAG Locality

This ablation tests whether the gain comes from DAG-local evidence, rather than from using the same judge in a different input format. We use the Grok/MATH setting because it is a difficult case and gives a clear test of evidence organization. The verifier is fixed across all columns. Only the input organization and the scoring protocol are changed.

We compare four settings. Whole-lin gives the verifier the full solution in a linear format and asks for an instance-level judgment. Whole-str gives the same solution in a structured format, but still asks for a whole-solution judgment. Prefix-step verifies each step with all previous statements as evidence. It uses step-level checking, but it does not use the cited parent set. Step-FLOW verifies each step only against its cited parents and then aggregates the local labels with the flow rule.

The two rows compare different graph sources. Base DAG uses the parsed source citations from the generated structured output. High-fidelity DAG uses a stronger source-tracing procedure to construct parent sets. The generated solution and the verifier are otherwise unchanged.

*Table 7.* Same-verifier ablation on Grok/MATH (AP, %). Whole-lin/whole-str judge the full solution with linear or structured formatting; Prefix-step uses all previous statements as evidence; Step-FLOW uses cited parents and flow aggregation.

| Setting | Whole-lin | Whole-str | Prefix-step | Step-FLOW |
|---|---|---|---|---|
| Base DAG | 53.09 | 52.65 | 57.82 | **65.70** |
| High-fidelity DAG | 52.97 | 52.73 | 62.95 | **72.87** |

Table 7 shows two effects. First, structured formatting alone does not improve whole-trace judging. Whole-lin obtains 53.09 AP on the Base DAG row, and Whole-str obtains 52.65 AP. On the High-fidelity DAG row, the corresponding values are 52.97 and 52.73. These values are almost unchanged.

Second, local evidence organization matters. On the Base DAG row, Prefix-step improves the score to 57.82 AP. Step-FLOW further improves it to 65.70 AP. This is 7.88 AP points above Prefix-step. On the High-fidelity DAG row, Prefix-step reaches 62.95 AP, while Step-FLOW reaches 72.87 AP. This is 9.92 AP points above Prefix-step.

The comparison between the two rows also shows the effect of graph quality. Step-FLOW improves from 65.70 to 72.87 AP when the high-fidelity DAG is used. Prefix-step also improves, from 57.82 to 62.95 AP. In contrast, the two whole-trace baselines remain near 53 AP. This indicates that the benefit is not only due to a stronger judge or a structured transcript. It comes from giving the verifier local support evidence and then aggregating the resulting labels with the dependency graph.

## D.2. Ablation 2: Verifier Model Sensitivity

The flow score depends on local reliability estimates. This ablation tests how sensitive the pipeline is to the verifier model. We keep the graph, evidence protocol, label space, and deterministic post-processing fixed. We change only the verifier.

The main experiments use GPT-5.2 as the step verifier. Here we compare two additional verifiers. Qwen-3.5-9B is an open-source reasoning verifier. GPT-4o is a proprietary verifier that is not specialized for this local reasoning protocol. FLOW uses the standard graph for each setting. FLOW+ uses the high-fidelity graph. The Avg. column is the arithmetic mean over the three generators.

Table 8 shows that verifier choice has a large effect. On GSM8K, Qwen-3.5-9B gives strong results. Its FLOW average is 76.18 AP, and its FLOW+ average is 83.30 AP. FLOW+ is the stronger Qwen variant on the average row. It also gives the best values for Gemini and Grok, with 80.00 and 88.13 AP. The Qwen FLOW value is highest for GPT, with 84.72 AP.

The GPT-4o verifier is much weaker in the same evidence-only setting. On GSM8K, its FLOW average is 33.01 AP, and its FLOW+ average is 26.91 AP. The same pattern appears on MATH. Qwen FLOW+ reaches 70.05 AP on average, while

*Table 8.* Verifier-model sensitivity under the same SRM pipeline (AP, %). The main experiments use GPT-5.2; this table reports an open-source reasoning verifier (Qwen-3.5-9B) and a non-reasoning proprietary verifier (GPT-4o). Avg. is the arithmetic mean over GPT/Gem./Grok generators.

| Dataset | Judge | Variant | GPT | Gem. | Grok | Avg. |
|---------|-------|---------|-----|------|------|------|
| GSM8K | Qwen-3.5-9B | Flow | **84.72** | 72.84 | 70.99 | 76.18 |
| | Qwen-3.5-9B | Flow+ | 81.78 | **80.00** | **88.13** | **83.30** |
| | GPT-4o | Flow | 34.54 | 31.41 | 23.08 | 33.01 |
| | GPT-4o | Flow+ | 31.73 | 28.61 | 20.38 | 26.91 |
| MATH | Qwen-3.5-9B | Flow | 62.38 | 66.98 | 59.55 | 62.97 |
| | Qwen-3.5-9B | Flow+ | **78.41** | **68.87** | **62.87** | **70.05** |
| | GPT-4o | Flow | 35.52 | 35.27 | 25.29 | 32.03 |
| | GPT-4o | Flow+ | 28.09 | 43.09 | 23.75 | 31.64 |
| HumanEval | Qwen-3.5-9B | Flow | 73.69 | **70.01** | 64.43 | **69.38** |
| | Qwen-3.5-9B | Flow+ | **70.00** | 52.01 | **77.77** | 66.59 |
| | GPT-4o | Flow | 19.27 | 11.73 | 26.45 | 19.15 |
| | GPT-4o | Flow+ | 21.86 | 12.56 | 28.78 | 21.07 |

*Table 9.* Graph organization and aggregation comparison on the GPT generator (AP, %). AvgTok is the average prompt-token proxy for the graph-baseline runner.

| Dataset | Graph organization | Flow AP | MaxErr AP | AvgTok |
|---------|--------------------|---------|-----------|--------|
| GSM8K | SRM-DAG | 65.8 | 59.8 | 700 |
| | PARC | 58.4 | 56.1 | 3374 |
| | GoV proxy | **81.8** | 77.4 | 3535 |
| MATH | SRM-DAG | 63.4 | 54.4 | 968 |
| | PARC | **65.4** | 61.8 | 4334 |
| | GoV proxy | 62.6 | 58.5 | 5152 |

GPT-4o FLOW and FLOW+ obtain 32.03 and 31.64 AP. On HumanEval, Qwen remains much stronger than GPT-4o. The Qwen FLOW average is 69.38 AP, and the Qwen FLOW+ average is 66.59 AP. The GPT-4o averages are 19.15 and 21.07 AP.

The table therefore supports two conclusions. First, the SRM pipeline is not tied to one closed-source verifier, since Qwen-3.5-9B gives competitive results in several settings. Second, local verifier quality remains important, and stronger reasoning-oriented verifiers tend to perform better. Graph aggregation can organize useful local signals, but it cannot fully remove verifier noise. This is especially important for nodes that have large downstream influence.

## D.3. Ablation 3: Graph Organization and Source Quality

Graph structure affects the pipeline in two ways. It determines which evidence the verifier sees for each step. It also determines how local labels are propagated to the final score. This subsection studies graph organization and graph-source quality.

A sparse graph can omit necessary support and make a correct step look unsupported. A dense graph can introduce irrelevant evidence and make local verification noisier. The following two tables test these issues under controlled settings.

### D.3.1. GRAPH ORGANIZATION UNDER A FIXED GENERATOR

Table 9 compares three graph organizations under the same GPT generator. SRM-DAG is the parsed dependency graph used by the SRM pipeline. PARC is a premise-augmented graph baseline. GoV proxy applies the graph-of-verification protocol on a high-fidelity graph. The entity-graph control is omitted because its edges are not step-level support citations and therefore do not test the same SRM hypothesis.

For each graph organization, the table reports FLOW AP, MaxErr AP, and AvgTok. MaxErr uses the worst local step score as the instance score. AvgTok is an average prompt-token proxy for the graph-baseline runner.

On GSM8K, GoV proxy has the highest FLOW AP, with 81.8. SRM-DAG obtains 65.8, and PARC obtains 58.4. The AvgTok values differ strongly. SRM-DAG uses 700 tokens on average, while PARC and GoV proxy use 3374 and 3535.

This shows that stronger graph organization can improve AP, but it can also increase prompt cost.

On MATH, the ranking is different. PARC obtains the highest FLOW AP, with 65.4. SRM-DAG obtains 63.4, and GoV proxy obtains 62.6. The three values are close, but their token costs are not. SRM-DAG uses 968 AvgTok, while PARC uses 4334 and GoV proxy uses 5152.

The table also separates graph organization from aggregation. In all six rows, FLOW AP is higher than MaxErr AP under the same graph. The gains are 6.0, 2.3, and 4.4 AP points on GSM8K, and 9.0, 3.6, and 4.1 AP points on MATH. Thus, graph organization matters, but the aggregation rule also matters within each graph.

### D.3.2. GRAPH SOURCE UNDER A FIXED QWEN VERIFIER

In this section, we explore the impact of graph-construction quality on the verifier judgment. FLOW− uses the raw parsed graph. FLOW uses a graph constructed by Qwen-3-27B. FLOW+ uses a GPT-5.2 high-fidelity DAG. The verifier is Qwen-3.5-9B in all rows. This isolates the effect of the parent sets from the effect of the verifier.

*Table 10.* Graph-source comparison with a Qwen verifier (AP, %). FLOW− uses raw parsed graphs; FLOW uses Qwen-built graphs; FLOW+ uses GPT-5.2 high-fidelity DAGs. Avg. is the arithmetic mean over the three generators.

| Data | Model | FLOW− | FLOW | FLOW+ |
|---|---|---|---|---|
| GSM8K | GPT | **84.72** | 81.41 | 81.78 |
| | Gem. | 72.84 | 68.21 | **80.00** |
| | Grok | 70.99 | 80.70 | **88.13** |
| | Avg. | 76.18 | 76.78 | **83.30** |
| MATH | GPT | 62.38 | 68.23 | **78.41** |
| | Gem. | 66.98 | 65.05 | **68.87** |
| | Grok | 59.55 | 60.06 | **62.87** |
| | Avg. | 62.97 | 64.45 | **70.05** |

The GSM8K block shows that graph improvement is not monotone in every cell. For GPT, FLOW− obtains 84.72 AP, which is higher than FLOW and FLOW+. For Gemini and Grok, FLOW+ is best, with 80.00 and 88.13 AP. The average row is the most important summary. FLOW+ reaches 83.30 AP, compared with 76.18 for FLOW− and 76.78 for FLOW.

The MATH block gives a clearer pattern. FLOW+ is best for all three generators. It obtains 78.41 AP for GPT, 68.87 for Gemini, and 62.87 for Grok. The average improves from 62.97 under FLOW− and 64.45 under FLOW to 70.05 under FLOW+.

These results show that better parent sets usually improve the final score, especially on the average rows. The effect is not automatic for every generator. This is expected, because a graph can help only when it gives the verifier the right local evidence and avoids irrelevant context.

## E. Other Results

This subsection studies three practical questions. First, it asks whether structured prompting changes the base answer accuracy. Second, it tests whether aggregation behavior changes with reasoning length. Third, it checks whether perturbing high-visitation nodes has a stronger effect than perturbing low-visitation nodes.

### E.1. Effect of Structured Prompting on Answer Accuracy

Table 11 compares plain prompting with structured prompting. The goal is not to evaluate the detector directly. The goal is to test whether the structured format itself changes the generator's answer accuracy.

The table reports plain accuracy and structured accuracy for each generator. Avg. is the arithmetic mean over GPT, Gemini, and Grok. $\Delta$ is the structured average minus the plain average. This comparison uses zero-shot prompts for both formats, so it should be read as a prompting diagnostic rather than as a direct comparison with the few-shot main experiments.

The effect of structured prompting is task-dependent. On GSM8K, the plain average is 92.34 and the structured average is 91.63, giving $\Delta = -0.71$. GPT and Gemini decrease, while Grok slightly increases from 92.42 to 92.72. On MATH, the average also decreases, from 74.80 to 74.00. All three generators are slightly lower under the structured prompt.

*Table 11.* Plain vs. structured answer accuracy. Avg. is the arithmetic mean over GPT/Gem./Grok; $\Delta$ is Structured Avg. minus Plain Avg. This comparison uses zero-shot prompts for both formats and is therefore a prompting diagnostic rather than a direct comparison with the few-shot main experiments.

| Dataset | Plain GPT | Plain Gem. | Plain Grok | Plain Avg. | Struct. GPT | Struct. Gem. | Struct. Grok | Struct. Avg. | $\Delta$ |
|---|---|---|---|---|---|---|---|---|---|
| GSM8K | 90.90 | 93.71 | 92.42 | 92.34 | 90.14 | 92.04 | 92.72 | 91.63 | -0.71 |
| MATH | 68.70 | 82.20 | 73.70 | 74.80 | 67.30 | 81.60 | 73.10 | 74.00 | -0.80 |
| HotpotQA | 74.18 | 74.14 | 76.34 | 74.89 | 71.61 | 68.88 | 61.26 | 67.25 | -7.64 |
| HumanEval | 84.15 | 90.85 | 84.76 | 86.92 | 87.80 | 93.90 | 90.24 | 90.65 | +3.73 |

HotpotQA has the largest decrease. The average drops from 74.89 to 67.25, with $\Delta = -7.64$. The drop is especially large for Grok, from 76.34 to 61.26. This suggests that the structured format can hurt tasks where the trace is short and factual retrieval is central.

HumanEval shows the opposite pattern. The average increases from 86.92 to 90.65, with $\Delta = +3.73$. All three generators improve. GPT increases from 84.15 to 87.80. Gemini increases from 90.85 to 93.90. Grok increases from 84.76 to 90.24.

These results show that structured prompting does not uniformly improve base answer accuracy. Therefore, the detection gains in the main experiments should not be explained only by better generation. The structured format mainly provides explicit conditions, steps, and source citations for verification.

### E.2. Aggregation Behavior by Reasoning Length

Table 12 studies whether the aggregation rule behaves differently for short and long traces. We bucket examples by the number of reasoning steps. The table reports macro-averaged AP over usable dataset–model settings. The Settings column gives the number of usable settings in each bucket. The last column is $F-Max$, which equals FLOW AP minus MaxErr AP.

*Table 12.* Aggregation behavior by reasoning-length bucket. Values are macro-averaged AP(%) over usable dataset–model settings; $F-Max$ is Flow AP minus MaxErr AP.

| Steps | Settings | Flow | Mean | Max | $F-Max$ |
|---|---|---|---|---|---|
| 1–2 | 8 | 67.9 | 68.7 | 67.4 | 0.5 |
| 3 | 9 | 73.1 | 70.8 | 62.7 | 10.4 |
| 4 | 11 | 77.0 | 74.0 | 59.1 | 17.8 |
| 5 | 11 | 66.7 | 66.1 | 54.5 | 12.2 |
| 6 | 12 | 75.4 | 76.1 | 61.0 | 14.4 |
| 7 | 8 | 82.5 | 79.7 | 51.9 | 30.6 |
| 8+ | 11 | 81.2 | 84.4 | 58.9 | 22.3 |

For very short traces, the aggregation methods are close. In the 1–2 step bucket, FLOW obtains 67.9 AP, Mean obtains 68.7, and Max obtains 67.4. The $F-Max$ gap is only 0.5. In this regime, there are few dependencies to aggregate, so graph propagation has limited room to differ from simple local summaries.

For longer traces, the gap between FLOW and Max becomes larger. The gap is 10.4 for 3 steps, 17.8 for 4 steps, 12.2 for 5 steps, 14.4 for 6 steps, 30.6 for 7 steps, and 22.3 for 8 or more steps. FLOW is best in the 3-step, 4-step, 5-step, and 7-step buckets. Mean is best in the 1–2, 6-step, and 8+ buckets.

The table does not show that FLOW always increases with length. It shows a more specific pattern. MaxErr becomes less competitive as traces become longer, while FLOW remains strong in most length buckets. This supports the use of graph aggregation when a trace contains more intermediate dependencies.

### E.3. Node Influence and Visitation Probability

We flip one node label within each visitation-probability bin and report the resulting mean FLOW. The bins are ordered by visitation probability. Higher-$\rho$ bins correspond to nodes that are visited more often by dependency-respecting support chains.

This experiment is linked to the weighted Lipschitz interpretation of the flow score. A node with higher visitation probability has more opportunities to affect downstream mass. A local verifier error on such a node should therefore have a larger effect on the final score.

*Table 13.* Mean FLOW after flipping one node label within each visitation-probability bin. Values are in $[0, 1]$ and are macro-averaged over base models; higher-$\rho$ bins correspond to structurally more visited nodes.

| Data | $\rho_1$ | $\rho_2$ | $\rho_3$ | $\rho_4$ |
|---|---|---|---|---|
| GSM8K | 0.393 | 0.666 | 0.964 | 1.000 |
| MATH | 0.452 | 0.556 | 0.727 | 0.917 |
| HotpotQA | 0.594 | 0.599 | 0.864 | – |
| HumanEval | 0.213 | 0.266 | 0.338 | 0.621 |

The trend is clear on GSM8K and MATH. On GSM8K, the value increases from 0.393 in $\rho_1$ to 1.000 in $\rho_4$. On MATH, it increases from 0.452 to 0.917. HumanEval also increases across bins, from 0.213 to 0.621. HotpotQA has no reported $\rho_4$ value, and the first two bins are close, with 0.594 and 0.599. The third bin is higher, with 0.864.

These results support the structural interpretation of node importance. Errors on highly visited nodes are more influential because more dependency-respecting paths pass through them. This also suggests a practical extension. A system can spend more verification budget on high-$\rho$ nodes, where local mistakes are more likely to affect the final detection score.

## F. Regression-Based Quantification of DAG Utility

The preceding ablations evaluate hand-designed aggregation rules. Here we ask whether the dependency matrix itself contains predictive signal beyond the unordered list of local verifier labels.

**Dataset selection.** We focus on MATH. GSM8K has low error prevalence and shallow extracted graphs, making structure-induced gains unstable. MATH has higher error prevalence and richer dependency patterns, so it better isolates the incremental value of DAG-aware features.

**Problem setup.** For each solution we observe a step-level error vector $\hat{\mathbf{e}} = (\hat{e}_1, \dots, \hat{e}_S) \in \{0, 1\}^S$, where $\hat{e}_i = 1$ indicates that step $i$ is judged erroneous, and an adjacency matrix $A$ induced by Source citations. We compare structure-agnostic summaries of $\hat{\mathbf{e}}$ with DAG-aware features of $(\hat{\mathbf{e}}, A)$.

**Model and evaluation.** We fit regularized logistic regression models to predict the instance-level label $y \in \{0, 1\}$. Features are standardized, regularization is tuned by cross-validation, and evaluation uses AP under repeated stratified 10-fold resampling. Because positives are relatively rare, AP is more informative than ROC-AUC in this setting (Davis & Goodrich, 2006; Saito & Rehmsmeier, 2015).

*Table 14.* Regression-based DAG utility on MATH (AP, %). AGN.: structure-agnostic features from $\hat{\mathbf{e}}$ only. DAG: features using $(\hat{\mathbf{e}}, A)$. F: Full setting. NS: NoShortcut setting.

| FEAT. | GPT | GEM. | GROK | AVG. |
|---|---|---|---|---|
| AGN. (F) | 75.80 | 79.00 | 69.43 | 74.74 |
| DAG (F) | **77.37** | **80.70** | **74.60** | **77.56** |
| AGN. (NS) | 74.29 | 75.07 | 69.58 | 72.98 |
| DAG (NS) | **77.12** | **77.54** | **73.50** | **76.05** |

DAG-aware features improve average AP from 74.74 to 77.56 in the Full setting and from 72.98 to 76.05 in the NoShortcut setting. This supports the claim that the graph contributes complementary signal beyond structure-free summaries of local labels.

### F.1. Feature Definitions for Regression-Based Aggregation

Let $C$ be the number of conditions and $S$ the number of steps. Each instance is represented by step-level error indicators $\hat{\mathbf{e}} \in \{0, 1\}^S$ and an adjacency matrix $A \in \{0, 1\}^{S \times (C+S)}$, whose columns index condition nodes followed by step nodes. We write

$$A = \begin{bmatrix} A^{cs} & A^{ss} \end{bmatrix}, \qquad A^{cs} \in \{0, 1\}^{S \times C}, \quad A^{ss} \in \{0, 1\}^{S \times S}.$$

An entry $A^{ss}_{i,k} = 1$ means that step $i$ cites step $k$ as a parent.

**Label-only summaries.** We use $C$, $\mathbb{I}\{\sum_i \hat{e}_i > 0\}$, $\mathbb{I}\{\sum_i \hat{e}_i = 0\}$, $\hat{e}_S$, the normalized last-error position, and coarse prefix/suffix error rates.

**Edge-pattern features.** For step-to-step edges $E^{ss} = \{(k \to i) : A^{ss}_{i,k} = 1\}$, define

$$N_{ab} := \sum_{(k \to i) \in E^{ss}} \mathbb{I}\{\hat{e}_k = a, \hat{e}_i = b\}, \qquad a, b \in \{0, 1\}.$$

We include $N_{11}$, $N_{10}/(|E^{ss}| + \varepsilon)$, $N_{00}/(|E^{ss}| + \varepsilon)$, and the number of condition-to-step edges.

**Node-importance features.** We include in-degree, out-degree, fan-in/fan-out among erroneous nodes, top-out-degree error rate, reachability to the terminal answer, distance-to-answer summaries, longest-path depth, and source-usage features $s = A^\top \hat{e}$.

# G. Failure Cases

We manually inspected extreme outliers and found two recurring failure modes.

**Strong implicit premises and pragmatic semantics.** Some problems hinge on implicit constraints that are not stated as explicit premises and are therefore difficult to recover as `Conditions`. A representative GSM8K example is:

> *"Carlos is planting a lemon tree. The tree will cost $90 to plant. Each year it will grow 7 lemons, which he can sell for $1.5 each. It costs $3 a year to water and feed the tree. How many years will it take before he starts earning money on the lemon tree?"*

The phrase "starts earning money" refers pragmatically to the first year after break-even, rather than merely the first year when cumulative profit is non-negative. If this premise is absent from the extracted supports, evidence-only local verification can miss the intended interpretation.

**Termination conditions and truncated enumeration.** Another failure mode arises when a solution relies on an implicit termination criterion, often through brute-force enumeration, and the generation is truncated mid-process. A representative MATH-style example is:

> *"Find the largest integer below `100` that is divisible by both its tens digit and its ones digit."*

Models often enumerate cases but may be cut off before the concluding step. We conservatively flag unfinished reasoning as incorrect, which handles the symptom but does not solve the underlying difficulty: local step checking is not well suited to certifying global termination properties. Under our generation budget, this truncation-induced failure occurs in $418/5990$ MATH solutions and was not observed on GSM8K.

# H. Additional PR Curves

The main text shows two representative precision–recall curves. Here we provide the remaining panels. Continuous-valued methods are shown as thresholded PR curves whenever per-instance scores are available; binary or ternary judge outputs are shown as attainable operating points.

# I. Prompt Templates

This section summarizes the prompt templates used in the released pipeline. We show compact versions of the main mathematical prompts; the full templates, including all in-context exemplars and the HotpotQA/HumanEval variants, are released with the code. The GSM8K and MATH sampling prompts share the same Condition/Step/Source abstraction, but differ in demonstrations ($K = 6$ for GSM8K and $K = 8$ for MATH) and small formatting details. HotpotQA and HumanEval use task-specific variants: HotpotQA cites Wikipedia titles or previous steps, whereas HumanEval restates programming requirements and returns final code.

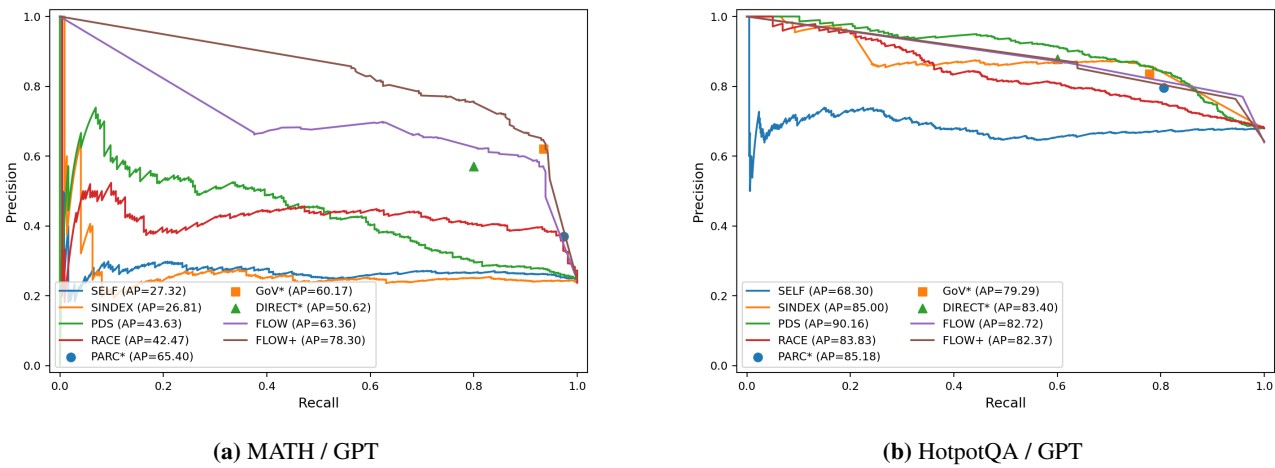

**(a)** MATH / GPT

**(b)** HotpotQA / GPT

*Figure 5.* Additional GPT-generator PR curves. GSM8K/GPT and HumanEval/GPT are shown in the main text.

**(a)** GSM8K / Gemini

**(b)** MATH / Gemini

**(c)** GSM8K / Grok

**(d)** MATH / Grok

*Figure 6.* Additional PR curves on the main math benchmarks for Gemini and Grok generators.

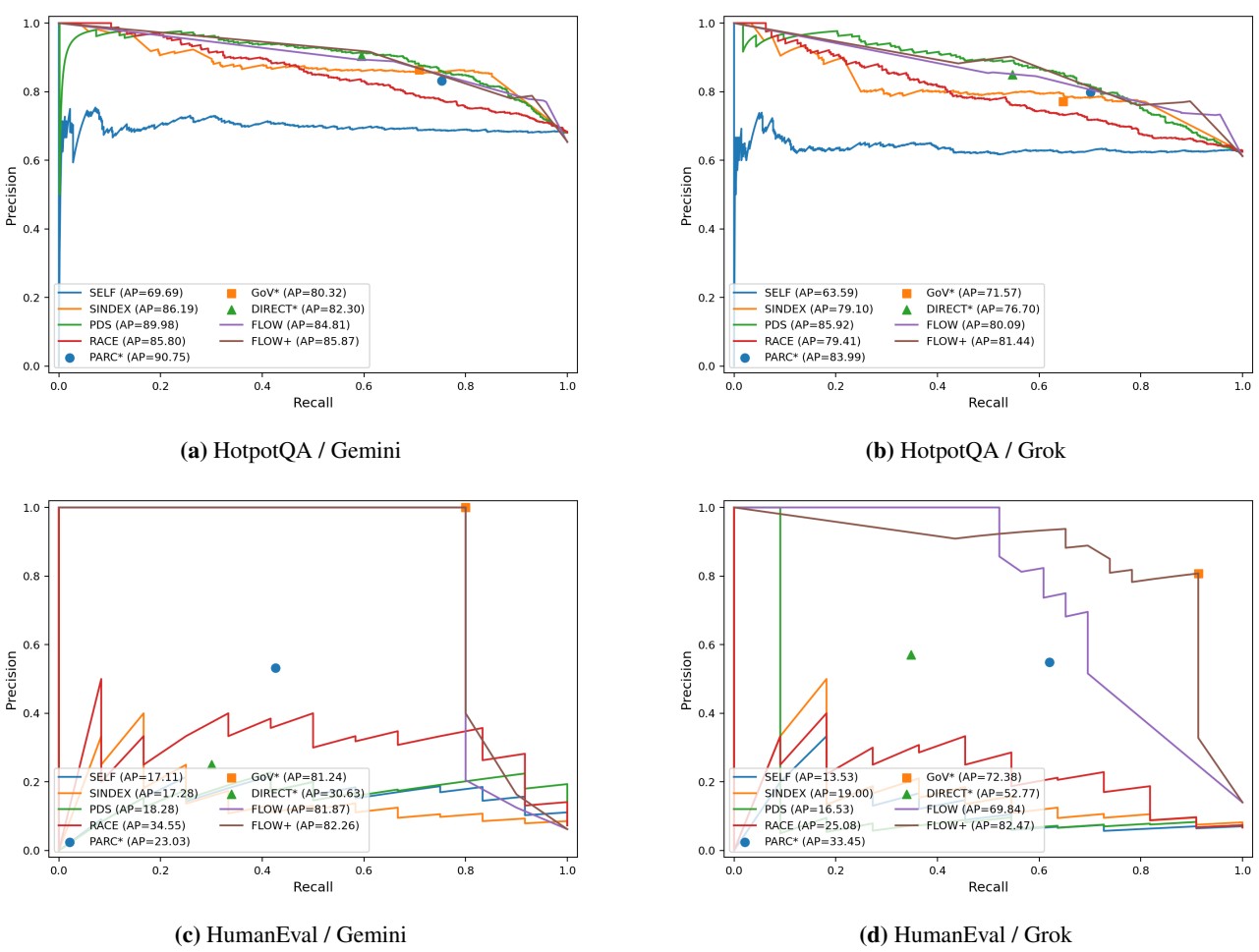

(a) HotpotQA / Gemini

(b) HotpotQA / Grok

(c) HumanEval / Gemini

(d) HumanEval / Grok

*Figure 7.* Additional transfer PR curves for the Gemini and Grok generators.

```
Structured-rationale sampling prompt (template)

[System]
You format a solution into atomic Conditions and sourced Steps.

[Task]
Follow STRICT FORMAT and STRICT RULES.

STRICT FORMAT

Conditions:
Condition1: ...
Condition2: ...
...

Steps:
Step1: Source: Condition{i} & Condition{j}; {ONE sentence}.
Step2: Source: Condition{i} & Step1; {ONE sentence}.
...

Final: Source: Condition{i} & Step{j}; Answer: {final answer}

STRICT RULES

A) Conditions
- Rewrite the problem into atomic factual statements as Condition1..Conditionm.
- Each Condition must be exactly ONE sentence.
- Conditions must be faithful to the question text; do not solve the problem
  and do not add derived facts.
- Include a Condition for what the question asks when useful for verifying
  the final answer.
- Conditions must be standalone and unambiguous.

B) Steps
- Use exact pattern: "Step{k}: Source: ...; {ONE sentence}."
- Each Step must be exactly ONE sentence.
- Source contains only node identifiers joined by " & ": Condition# and/or
  earlier Step#.
- Source must cite the minimal prior Conditions and Steps needed to verify
  the current Step.
- Do NOT copy full source sentences into Source.
- Do NOT cite future Steps.

C) Final
- Use exact pattern: "Final: Source: ...; Answer: {final answer}".
- Final Source contains only Condition# and/or Step# identifiers joined by " & ".
- Output only Conditions, Steps, and Final after "A:".

Question: <problem text>
A:
```

*Figure 8.* Compressed structured-rationale sampling prompt. The released MATH prompt uses this Condition/Step/Final format with $K = 8$ demonstrations; the GSM8K prompt uses the same source-tag abstraction with $K = 6$ demonstrations and minor formatting differences. Full prompts are included in the released code.

---

**Step-level evidence-only verifier**

```
[System]
You are a strict logic checker and arithmetic checker.
Verify whether the Target statement is mathematically correct given the Evidence.

Semantic labels:
- 0 = correct / entailed
- 0.5 = insufficient information / ambiguity
- 1 = incorrect / contradicted / unsupported

Rules:
1) Recompute numeric and algebraic claims independently.
2) Output 1 if arithmetic is wrong, logic is wrong, or the target contradicts
   the evidence.
3) Output 0 only if the target is entailed by the evidence and the arithmetic
   is correct.
4) Use Background only to resolve references and ordering; do not use the gold
   answer.
5) If the target is incomplete, output 1.

Output rule:
First write a concise Reason that checks the evidence and any arithmetic.
Then write Final label: on its own line, followed by exactly one label on the
last line:
0
or
0.5
or
1

Do not put any other text on or after the final label line.

Background:
{question}

Evidence:
{source}

Target statement:
{cot}
```

*Figure 9.* Evidence-only step verifier prompt. The label is an error score $e_t$; the flow reliability is $\hat{q}_t = 1 - e_t$.

---

**Direct whole-solution judge (baseline)**

```
[System]
You are a strict evaluator for reasoning outputs.
Judge whether the model solution should be treated as wrong / hallucinated.
Output 1 if the final answer is wrong, the reasoning contains decisive
unsupported claims, or the trace should be treated as hallucinated.
Output 0 only if the solution is acceptable overall.

Question:
{question}

Model solution:
{solution}

Return one label on the last line: 0 or 1.
```

*Figure 10.* Direct whole-solution judge prompt used for the holistic judging baseline.

---

**Math answer-equivalence judge**

```
Given a math problem, a reference answer, and a model answer, decide whether
the two answers are mathematically equivalent.
Return JSON only: {"equivalent": true/false, "reason": "short reason"}.
Do not judge the reasoning trace; only judge final-answer equivalence.
```

*Figure 11.* Prompt used for final-answer equivalence checks in mathematical tasks.

High-fidelity DAG source tracing

```
[System]
You construct parent sets for a reasoning DAG.
Given conditions, previous steps, and the current step, output the minimal
parent set needed to support the current step.

Return only identifiers joined by " & ".
Valid identifiers are Condition1..ConditionC and Step1..Step(i-1).
Never cite future steps. Do not explain your answer.
```

*Figure 12.* Source-tracing prompt used by FLOW+ to construct high-fidelity parent sets.

