# OpenReview forum: "Hallucination Detection from Structural Reasoning Model"
_ICML.cc/2026/Conference — ICML 2026 regular_

### Official Review · Reviewer_uiba · 2026-02-26

**Soundness:** 3
**Presentation:** 2
**Significance:** 2
**Originality:** 3
**Overall Recommendation:** 4
**Confidence:** 4

**Summary:**

The authors propose a Structural Reasoning Model, which leverages specific prompts to induce the model to output structured reasoning steps, and then integrates them into a corresponding DAG. By performing local verification on each step (i.e., each vertex in the graph except for conditions), the correctness probability is obtained. Finally, the Flow Score is computed for hallucination detection. The authors explain the meaning of the Flow Score from an information-theoretic perspective and demonstrate in experiments that the proposed method achieves higher Average Precision (AP) than baselines.

**Compliance With Llm Reviewing Policy:**

Affirmed.

**Final Justification:**

I believe my concerns have been fully addressed. The authors have demonstrated the robustness and computational efficiency of their method through further experiments, and have objectively discussed the impacts of prompting and graph accuracy. Consequently, I will raise my score to 4.

**Key Questions For Authors:**

1.	To what extent does the structured reasoning prompt affect accuracy? Could it potentially reduce the original reasoning ability due to the requirement of generating structured output? Does the method heavily rely on the model’s ability to produce the correct structure (e.g., the difference in performance between FLOW and FLOW+)?
2.	Is the Flow Score sensitive to the number of reasoning steps? Does it drop significantly when the number of steps increases?
3.	Is the Flow Score sensitive to misjudgments at individual nodes? Do such errors accumulate? As suggested in Corollary 4.3, does this effect become more pronounced for nodes with larger $\rho_u$？
4.	In Section 5.3, regarding step-level verification, it is mentioned that “if the primary judge predicts incorrectly or reports low confidence, we re-evaluate…” However, later it is stated that a temperature of 0.0 is used, meaning the results of two decodings should be exactly the same. Doesn’t that make the re-evaluation effectively ineffective?
5.	What is the computational efficiency of the Flow Score compared to other baselines, since it requires multiple forward passes through the LLM for local verification.

**Limitations:**

yes

**Strengths And Weaknesses:**

Strengths:

1.	Constructing the CoT into a corresponding DAG effectively mitigates the bias that can arise when evaluating long CoTs, allowing for a thorough assessment of detailed steps.
2.	Theoretical analysis demonstrates that the Flow Score corresponds to the proportion of task-relevant content that is erased and the degree of mutual information loss, providing strong interpretability for the metric.
3.	Experimental results show that the performance of FLOW and FLOW+ surpasses other baselines.

Weaknesses:

1.	The method requires the use of specific prompts to generate structured outputs, which is not ideal for real-world applications, especially when users need to specify such prompts. Furthermore, whether forcing the model to output structured reasoning processes will impact its effectiveness remains unknown.
2.	As the reasoning steps become longer, the Flow Score may become very small due to the increase in steps, and there is a lack of theoretical or experimental evidence regarding its stability in such scenarios.
3.	Inaccuracies or errors in step-level verification may lead to instability in the overall Flow Score, as these errors can propagate along the graph and affect all subsequent nodes.

---

> ### Author Rebuttal · Authors · 2026-03-31
>
> ## Reviewer uiba
>
> Abbrev. G8=GSM8K, M=MATH, HQ=HotpotQA, HE=HumanEval. GPT=GPT-4o-mini, Gem=Gemini-2.5-Flash-Lite, Grok=Grok-4-Fast. FLOW+=Golden-DAG variant.
>
> **1. The influence of structured prompting and the correctness of the graph.**
> Thanks. Structured prompting is task-dependent: it helps on HE, hurts on HQ, and is usually modest on G8/M. We will revise the paper to make this tradeoff explicit instead of implying uniform benefit.
>
> |Data|Model|Plain Acc|Struct. Acc|
> |---|---|--:|--:|
> |G8|GPT|90.90|90.14|
> |G8|Gem|93.71|92.04|
> |G8|Grok|92.42|92.72|
> |HE|GPT|84.15|87.80|
> |HE|Gem|90.85|93.90|
> |HE|Grok|84.76|90.24|
> |M|GPT|68.70|67.30|
> |M|Gem|82.20|81.60|
> |M|Grok|73.70|73.10|
> |HQ|GPT|74.18|71.61|
> |HQ|Gem|74.14|68.88|
> |HQ|Grok|76.34|61.26|
>
> _Table. Plain vs. structured answer accuracy._
>
> To study graph correctness, we also built a higher-quality Golden-DAG with a local Qwen-3.5-9B pipeline and evaluated FLOW and FLOW+ with the same Qwen verifier:
>
> |Data|Model|FLOW-|FLOW|FLOW+|
> |---|---|--:|--:|--:|
> |G8|GPT|84.70|81.41|81.80|
> |G8|Gem|72.80|68.21|80.00|
> |G8|Grok|71.00|80.70|88.10|
> |M|GPT|62.40|68.23|78.40|
> |M|Gem|67.00|65.05|68.90|
> |M|Grok|59.60|60.06|62.90|
>
> _Table. Qwen verifier: raw-DAG FLOW- vs. Qwen-built FLOW (AP, %)._ vs. GPT-5.2-built FLOW+
>
> These results show that graph quality matters. In most settings, better graphs improve final FLOW, sometimes clearly. The effect is not perfectly uniform, but the overall pattern is that better graph structure usually improves aggregation quality. We will add this comparison and discussion.
>
> **2. Sensitivity to longer reasoning chains.**
> Thanks. We agree there is a real length effect: on some benchmarks, especially G8 and M, mean FLOW becomes less favorable as CoTs get longer. We therefore added a bucketed analysis by effective reasoning length.
>
> We first report mean FLOW over four buckets (1--4, 5--7, 8--10, 11+). The trend mainly appears on the harder benchmarks, and is weak or mixed on HQ and HE.
>
> |Benchmark\Bucket|1--4|5--7|8--10|11+|
> |---|--:|--:|--:|--:|
> |G8|0.073|0.079|0.131|0.562|
> |HQ|0.611|0.611|0.518|--|
> |HE|0.112|0.082|0.152|0.117|
> |M|0.168|0.218|0.290|0.443|
>
> _Table. Mean FLOW by length bucket._
>
> However, this should not be read as a purely mechanical failure from having more steps. Longer CoTs are also more likely to be incorrect, especially on G8 and M, supporting the chain longer CoT -> harder problem -> higher error/hallucination rate. We also observe that direct whole-trace judging becomes less reliable on longer CoTs, with a clearly rising error rate where direct-judge coverage is available:
>
> |Benchmark\Bucket|1--4|5--7|8--10|11+|
> |---|--:|--:|--:|--:|
> |G8 error rate|5.92%|8.69%|12.30%|33.33%|
> |G8 DirectJudge error|5.16%|7.32%|8.99%|50.00%|
> |HQ error rate|63.64%|63.02%|44.44%|--|
> |HE error rate|10.71%|10.75%|10.58%|12.00%|
> |M error rate|16.31%|19.12%|26.18%|36.58%|
> |M DirectJudge error|19.54%|29.47%|40.85%|47.59%|
>
> _Table. Error-related statistics by length bucket; DirectJudge error where available._
>
> So yes, longer chains are harder, but not uniquely for FLOW: both the underlying error rate and direct whole-trace judging worsen. This is exactly where step-wise verification is most motivated. For very hard problems or very long CoTs, a natural extension is hierarchical steps, so verification can operate at multiple resolutions. We will add this to the revision.
>
> **3. Sensitivity to node-level misjudgments / larger $\rho_u$.**
> We add a direct perturbation analysis: for each example, we flip one node label within an importance bin and measure the change in FLOW. Sensitivity increases clearly with node importance, and the trend is approximately linear overall, consistent with Corollary 3.3. Averaging $\Delta$FLOW over the three non-Golden models for each task gives:
>
> |Data|$\rho=1$|$\rho=2$|$\rho=3$|$\rho=4$|
> |---|--:|--:|--:|--:|
> |G8|0.455|0.626|0.825|0.937|
> |M|0.186|0.275|0.444|0.686|
> |HQ|0.145|--|0.208|0.408|
> |HE|0.156|0.157|0.233|0.567|
>
> _Table. Mean $\Delta$FLOW after flipping one node label in each $\rho$ bin._
>
> This is also practically useful: it suggests more computation for structurally important nodes, e.g., a stronger judge only on high-$\rho$ nodes.
>
> **4. Temperature 0.0 and the cascade.**
> The second pass uses an independently deployed verifier, not a re-decode of the same model. So temperature 0.0 makes each verifier deterministic, but does not make the cascade redundant.
>
> **5. Computational efficiency.**
> Our method does require multiple local verification calls, but FLOW itself is light once step labels are available: one DAG traversal propagates and aggregates signals. The extra cost therefore comes from step-wise verification, not FLOW aggregation. End-to-end, we agree that the current pipeline is more expensive than direct whole-trace judging; please see Reviewer WcCe, Point 1, for the detailed cost table comparing DirectJudge, our method, and sampling-based baselines on G8/M.

---

> > ### Author Rebuttal · Reviewer_uiba · 2026-04-04
> >
> > I thank the authors for their rebuttal. I believe my concerns have been fully addressed. The authors have demonstrated the robustness and computational efficiency of their method through further experiments, and have objectively discussed the impacts of prompting and graph accuracy. Consequently, I will raise my score to 4.

---

> > > ### Author Response · Authors · 2026-04-08
> > >
> > > Thank you very much for your thoughtful follow-up, for carefully revisiting our rebuttal, and for raising the score. We are very grateful that you found your concerns fully addressed.
> > >
> > > We especially appreciate your recognition that the additional analyses helped clarify the robustness and computational efficiency of our method, as well as the effects of structured prompting and graph accuracy. Your comments were very helpful in pushing us to make these points much clearer and more balanced.
> > >
> > > In the revision, we will incorporate these additions directly into the paper, including:
> > >
> > > (1) the plain-vs.-structured prompting comparison and the corresponding discussion of when structured prompting helps or hurts;
> > >
> > > (2) the graph-quality comparison across raw-DAG, Qwen-built DAG, and Golden-DAG variants;
> > >
> > > (3) the length-bucket and node-perturbation analyses on Flow sensitivity and robustness;
> > >
> > > (4) the clarification on the deterministic multi-verifier cascade and the computational-cost discussion.
> > >
> > > We will also revise the presentation so that these tradeoffs and limitations are stated more explicitly and objectively in the final version.
> > >
> > > Thank you again for your careful reading, constructive feedback, and helpful suggestions. We sincerely appreciate your engagement with our work.

---

### Official Review · Reviewer_WcCe · 2026-03-08

**Soundness:** 3
**Presentation:** 3
**Significance:** 3
**Originality:** 3
**Overall Recommendation:** 5
**Confidence:** 3

**Summary:**

This paper studies hallucination/error detection for large language models in multi-step reasoning tasks. To address the limitation that chain-of-thought (CoT) is typically presented as a linear sequence—making step dependencies implicit and unsupported intermediate claims hard to localize—the authors propose a Structural Reasoning Model (SRM). Using structured prompting, they extract “conditions–steps–source citations” from model outputs to build a dependency directed acyclic graph (DAG). Each step is locally verified against its directly cited evidence to produce step-level reliability signals, which are then propagated and aggregated on the DAG via a FlowScore to yield an instance-level risk score, with an information-theoretic interpretation (as the survival/loss of information along dependency chains). Experiments on GSM8K and MATH against sampling-consistency and LLM-as-judge baselines show consistent gains, and further analyses quantify the contributions of dependency structure and citation quality to performance.

**Compliance With Llm Reviewing Policy:**

Affirmed.

**Final Justification:**

The authors' rebuttal has addressed my main concerns; therefore, I have increased my score from 4 to 5.

**Key Questions For Authors:**

1. Cost (time/money).
Please report the practical cost of the proposed pipeline: how many model calls per instance (structured generation + per-step verification + any cascading re-checks), the average number of steps (T), and total token usage. Also provide end-to-end latency and estimated cost (or relative cost), and compare against sampling baselines (K samples) and whole-trace judging.
2. GPT-5.2 is strong: are the gains mainly from the judge?
Since GPT-5.2 is used for step verification, please report how well GPT-5.2 alone performs when directly judging the entire rationale/structured solution (single pass scoring or decision). How large is the gap to FLOW/FLOW+? This would clarify the contribution of the strong judge vs. the structured aggregation.
3. Baseline fairness: are the LLMs consistent?
Please clarify whether all baselines use the same base LLMs (for generating solutions) and the same judge/verifier models. If they differ, how do you rule out that performance differences are due to model strength rather than the method? Ideally, include controlled comparisons using the same judge (e.g., GPT-5.2).
4. Missing direct comparisons to graph/structured reasoning evaluation methods.
The paper is conceptually related to graph/structured reasoning work, but experiments mainly compare sampling- and judge-based baselines. Could you add at least one reproducible “graph-structured” baseline (e.g., simple graph aggregations such as max/mean/min or weakest-link-on-path)? If such methods are not comparable, please explain why (e.g., missing citation edges/output interfaces), to better locate where SRM’s gains come from.

**Limitations:**

yes

**Strengths And Weaknesses:**

**Soundness**

Overall, the work is technically complete with a clear, closed-loop methodological pipeline. It is supported by fairly comprehensive experimental comparisons and additional analyses, and it provides an interpretable theoretical framework. The conclusions are largely consistent with the presented evidence.

**Presentation**

The overall writing and structure are generally clear and well organized.

**Significance**

This work addresses an important and practical problem and advances the direction of “structured, localizable” hallucination detection in terms of methodology.

**Originality**

The idea of representing the reasoning process as a tree/graph (e.g., ToT/GoT) and performing local evidence verification or process-level evaluation on a graph structure has already been explored in a number of mature and closely related works. The more likely contribution of this paper lies in leveraging Source citations in the model output to directly construct a dependency DAG, conducting evidence-only step-by-step verification on top of it, and aggregating risk over the graph with a FlowScore that is more sensitive to shared supports. It further provides an “mutual-information loss” interpretation and a sensitivity analysis, making the phenomenon that shared premises matter more and errors accumulate along dependencies more intuitive and easier to analyze.

---

> ### Author Rebuttal · Authors · 2026-03-31
>
> **1. Cost / practicality.**
> Thank you for raising this important point. We summarize the average **prompt-token cost per example** by component below, separating **graph construction**, **verifier cost**, and **generation / sampling cost**. We compare **DirectJudge**, **our method**, and a **5-sample baseline**. Blanks mean *not applicable* or *not yet available*.
>
> | Dataset | Cost component | DirectJudge | Ours | Sample-5 |
> | ------- | -------------- | ----------: | ---: | -------: |
> | GSM8K   | Golden-DAG     |           — | 2114 |        — |
> | GSM8K   | Verifier       |         365 | 1797 |        0 |
> | GSM8K   | Generation     |         763 | 1522 |     2852 |
> | MATH    | Golden-DAG     |           — | 3142 |        — |
> | MATH    | Verifier       |         481 | 3143 |        0 |
> | MATH    | Generation     |        2814 | 1660 |    14070 |
>
> *Table. Average prompt-token cost per example.*
> We agree that our current pipeline is **more expensive than direct whole-trace judging** in its current form. The trade-off is that it produces **stepwise labels**, not just a final example-level score. These fine-grained labels are useful beyond the final SRM decision itself, and can support other downstream uses such as **stepwise judging / supervision**. This is closely related to the motivation of **process reward models (PRMs)**, where intermediate-step supervision is treated as a valuable signal in its own right Lightman et al. ICLR 2024.
> **2. Are the gains mainly from the judge?**
> Our same-judge ablation shows that the gain is not mainly from using a stronger judge: under a fixed verifier, local step-wise **FLOW** substantially outperforms both whole-trace judging and dense-prefix controls. Please see Reviewer pGVm, Point 2, for the full results and discussion.
>
> **3. Baseline fairness.**
> For judge-based baselines (e.g., direct whole-trace judging), we clarify that we use the same verifier family as in our method, **GPT-5.2**, so those comparisons are judge-matched. We also now include a weaker open-source verifier experiment with **Qwen-3.5-9B**, which shows that the overall performance does **not** drop dramatically under a weaker judge and that the method remains strong even with a fully local verifier. Please see **Reviewer pGVm, Point 2** for the corresponding open-source-verifier results and discussion.
>
> By contrast, sampling-based baselines do not rely on a separate judge model; their signal comes from cross-sample semantic consistency, so there is no additional judge to match in that case.
>
> **4. Missing graph-structured baselines.**
> We now include several simple graph-structured baselines on the same step-level verifier outputs, including **WeakLink**, **MaxErr**, **MeanErr**, and **InvFlow**. These new comparisons show that **FLOW** remains strongest on most GSM8K and HotpotQA settings and stays near the top on MATH and HumanEval, which more cleanly isolates the contribution of the aggregation rule itself. **We will add these baseline comparisons and the corresponding discussion to the revised paper.** Please see **Reviewer U2nR, Point 2** for the full table and discussion.

---

> > ### Author Rebuttal · Reviewer_WcCe · 2026-04-01
> >
> > Thank you to the authors for the detailed response. However, I still have a remaining concern regarding Comment 4, “Missing direct comparisons to graph/structured reasoning evaluation methods.” From my reading, the added response mainly demonstrates the advantage of the particular structure designed in your own method. My original intention, however, was to encourage comparisons with more independent graph- or structure-based baselines proposed in prior work, similar in spirit to methods such as SelfCheckGPT and SINDEX. I believe such comparisons would better position the proposed approach relative to existing structured reasoning or graph-based evaluation methods.

---

> > > ### Author Response · Authors · 2026-04-08
> > >
> > > Thank you for the helpful follow-up. To better localize where SRM’s gains come from, we added three structure-based baselines, plus graph structure ablation and aggregation ablation analyses; all will be included in the revision.
> > >
> > > **Structure-based baselines**
> > >
> > > - **PARC** (Mukherjee et al., ICML 2025) rewrites a linear chain into a premise DAG using minimal supporting premises for each step.
> > > - **GraphEval** (Sansford et al. KiL 2024 workshop) builds a graph from question entities and relations and verifies edges independently.
> > > - **GoV** (Fang et al., AAAI 2026) verifies graph-ordered node blocks against predecessor evidence in a DAG; because its graph-construction code is unavailable and block construction is left unspecified, we apply its verification stage on our best **Golden-DAG**.
> > >
> > >  For PARC and GraphEval, the primary focus is step-level verification accuracy, so we adopt an “any error renders the whole solution incorrect” rule, implemented via **MaxErr** aggregation. For GoV, verification stops once the first error is detected, which is consistent with **MaxErr** aggregation.
> > >
> > > Using GPT-4o-mini on GSM8K/MATH with the same judge and prompt as PARC, the main changed variable is graph organization/aggregation, so these serve as structure controls.
> > >
> > > | Data  | DAG      | PARC  | GoV       | GraphEval |
> > > | ----- | --------- | ----- | --------- | --------- |
> > > | GSM8K | 0.738     | 0.561 | **0.774** | 0.246     |
> > > | MATH  | **0.637** | 0.618 | 0.585     | 0.100     |
> > >
> > > These results suggest that structure quality can influence local judgments: on simpler GSM8K DAGs, it may have a larger effect, whereas on longer and more complex MATH CoTs, the denoising benefit of FLOW becomes more pronounced.
> > >
> > > We also computed **FLOW AP** and **MaxErr AP** for all methods. GraphEval is omitted because its graph does not support FLOW and is not well matched to our task.
> > >
> > > |Method|Data|Flow AP|MaxErr AP|AvgTok|
> > > |---|---|---|---|---|
> > > |DAG|GSM8K|0.738|0.665|1401.20|
> > > |PARC|GSM8K|0.584|0.561|3374.47|
> > > |GoV|GSM8K|0.818|0.774|3534.82|
> > > |DAG|MATH|0.637|0.580|1971.03|
> > > |PARC|MATH|0.654|0.618|4334.16|
> > > |GoV|MATH|0.626|0.585|5152.26|
> > >
> > > In all six settings, FLOW AP exceeds MaxErr AP under the same graph, showing that aggregation matters within each structure. Our pipeline is also competitive with or stronger than these baselines while using far fewer tokens.
> > >
> > > To further isolate SRM’s gains, we added two ablation studies, showing respectively the effect of the DAG under fixed aggregation and the effect of the aggregation under a fixed DAG.
> > >
> > > **Different DAG structure fixing FLOW**
> > >
> > > |Perturbation|GSM8K|MATH|
> > > |---|---|---|
> > > |Original DAG|0.6424|0.5591|
> > > |Drop 10% Edges|0.5126|0.4486|
> > > |Drop 20% Edges|0.4067|0.3438|
> > > |Drop 50% Edges|0.2317|0.2119|
> > > |Add 10% Edges|0.6392|0.5438|
> > > |Add 20% Edges|0.6368|0.5374|
> > > |Add 50% Edges|0.6349|0.5296|
> > > |Drop Condition Edges|0.0746|0.2112|
> > >
> > > As shown in table, dropping edges consistently reduces AP, with the largest drop from removing condition edges. Adding random edges is less harmful but still degrades AP.
> > >
> > > **Different aggregation fixing DAG**
> > >
> > > | Data | Model | FLOW  | InvFlow | MinErr | MaxErr | MeanErr | WeakLink |
> > > | ---- | ----- | ----- | ------- | ------ | ------ | ------- | -------- |
> > > | G8   | Gem   | 0.743 | 0.712   | 0.143  | 0.484  | 0.623   | 0.547    |
> > > | G8   | GPT   | 0.658 | 0.648   | 0.096  | 0.598  | 0.645   | 0.460    |
> > > | G8   | Grok  | 0.655 | 0.648   | 0.180  | 0.535  | 0.824   | 0.619    |
> > > | M    | Gem   | 0.727 | 0.711   | 0.318  | 0.543  | 0.752   | 0.560    |
> > > | M    | GPT   | 0.634 | 0.609   | 0.268  | 0.489  | 0.648   | 0.349    |
> > > | M    | Grok  | 0.657 | 0.651   | 0.206  | 0.540  | 0.620   | 0.508    |
> > > | HQ   | Gem   | 0.848 | 0.848   | 0.658  | 0.759  | 0.829   | 0.795    |
> > > | HQ   | GPT   | 0.827 | 0.827   | 0.648  | 0.757  | 0.818   | 0.781    |
> > > | HQ   | Grok  | 0.801 | 0.801   | 0.611  | 0.725  | 0.805   | 0.733    |
> > > | HE   | Gem   | 0.819 | 0.801   | 0.062  | 0.259  | 0.758   | 0.718    |
> > > | HE   | GPT   | 0.777 | 0.792   | 0.121  | 0.518  | 0.773   | 0.737    |
> > > | HE   | Grok  | 0.698 | 0.705   | 0.178  | 0.397  | 0.669   | 0.626    |
> > >
> > > As shown in table, **FLOW** achieves the best or near-best performance in most settings. Its advantage is especially clear on **GSM8K** and **HotpotQA**, and remains competitive on **MATH** and **HumanEval**. This pattern indicates that, even when the evidence structure is fixed, the way local verification signals are combined can substantially affect the final estimate.
> > >
> > > Overall, these results support a simple picture: **structure determines what each local check sees, and aggregation determines how well the remaining local noise is combined into the final score**.

---

### Official Review · Reviewer_U2nR · 2026-03-13

**Soundness:** 2
**Presentation:** 2
**Significance:** 2
**Originality:** 3
**Overall Recommendation:** 4
**Confidence:** 4

**Summary:**

When current model do reasoning, the CoT may hallucinate and generate wrong reasoning steps. In this paper the authors proposed a new method to better detect hallucation, which is called structural reasoning model. This model convert chain-of-thought (CoT) as a dependency DAG instead of a linear trace, verify each step against its cited parents, and then aggregate step-level errors with a flow-based score to detect hallucations. Then, the authors conducted experiments on GSM-8K and MATH and show better performance than sampling and judge-based baselines.

**Compliance With Llm Reviewing Policy:**

Affirmed.

**Final Justification:**

In the final review, I raise my score from 3 to 4 for a lot of experiments added in the new reply.

**Key Questions For Authors:**

See Strengths and Weaknesses

**Limitations:**

See Strengths and Weaknesses

**Strengths And Weaknesses:**

Strengthts
1. The paper shows better performance than two baselins of sampling and judge-based baselines.

Weaknesses:
1. Soundness 1: The paper only show the performance on math benchmarks, However, the halluciation detection should be a broad topic on a lot of tasks.
2. Soundness 2: The paper only conduct two experiments. One is the main evaluation in Section 6.1, another one is an ablation studies in Sectioin 6.2. The abaltion study appears somewhat limited. The paper can discuss more about the DAG construction method and aggregation stages. How do different DAG construction and aggregation methods affect the final performance
3. Significance: The idea seems very trivial. Turning a chain of thought to graph obviously can improve the performance.

---

> ### Author Rebuttal · Authors · 2026-03-31
>
> Abbrev. G8=GSM8K, M=MATH, HQ=HotpotQA, HE=HumanEval. GPT=GPT-4o-mini, Gem=Gemini-2.5-Flash-Lite, Grok=Grok-4-Fast. FLOW+=Golden-DAG variant.
>
> **1. Scope.**
> Thanks for the question. We added new HE and HQ results, which directly broaden the evaluation beyond math and show that the method transfers beyond G8/M. Please see Reviewer pGVm, Point 1, for the full results and discussion.
>
> **2. DAG construction and aggregation.**
> Thanks for the question. We have added rebuttal ablations to separate DAG construction from aggregation more clearly.
>
> On the DAG side, neither extreme is ideal. If the graph is too sparse, many correct steps appear unsupported because the verifier is given too little evidence, leading to many false error judgments. If the graph is too dense, irrelevant prefix context introduces noise. To make this concrete, we added a same-verifier ablation comparing Base DAG vs. Gold DAG under matched scoring protocols; please see Reviewer pGVm, Point 2. That table isolates the role of graph quality and shows more clearly what was missing in the original draft. The table in Reviewer uiba, Point 1 also show that the correctness of DAG also matters.
>
> On the aggregation side, following Reviewer WcCe, we added several simple graph-structured baselines on the raw parsed DAG while holding the same step-level verifier outputs fixed. We additionally report MinErr, MaxErr, MeanErr, and WeakLink. Across benchmarks, FLOW remains strongest on most G8 and HQ settings and stays near the top on M and HE, suggesting that the benefit is not tied to one ad hoc aggregator and that structure-aware aggregation is already useful even on noisy parsed DAGs.
>
> |Data|Model|FLOW|InvFlow|MinErr|MaxErr|MeanErr|WeakLink|
> |---|---|--:|--:|--:|--:|--:|--:|
> |G8|Gem|0.743|0.712|0.143|0.484|0.623|0.547|
> |G8|GPT|0.658|0.648|0.096|0.598|0.645|0.460|
> |G8|Grok|0.655|0.648|0.180|0.535|0.824|0.619|
> |M|Gem|0.727|0.711|0.318|0.543|0.752|0.560|
> |M|GPT|0.634|0.609|0.268|0.489|0.648|0.349|
> |M|Grok|0.657|0.651|0.206|0.540|0.620|0.508|
> |HQ|Gem|0.848|0.848|0.658|0.759|0.829|0.795|
> |HQ|GPT|0.827|0.827|0.648|0.757|0.818|0.781|
> |HQ|Grok|0.801|0.801|0.611|0.725|0.805|0.733|
> |HE|Gem|0.819|0.801|0.062|0.259|0.758|0.718|
> |HE|GPT|0.777|0.792|0.121|0.518|0.773|0.737|
> |HE|Grok|0.698|0.705|0.178|0.397|0.669|0.626|
>
> _Table. Aggregation comparison on the raw parsed DAG. MaxErr=worst local step score; MeanErr=average local score; WeakLink=weakest support along answer-reaching paths; InvFlow=reverse flow propagation from the answer node._
>
> **3. On significance / “the idea seems trivial.”**
> Thanks for the question. We agree that representing a CoT as a graph may sound simple at a high level, so the real issue is whether it yields a nontrivial empirical gain on realistic hallucination detection, rather than being only a cosmetic reformulation.
>
> Our new transfer results suggest that the benefit is task-dependent rather than automatic. As discussed in Reviewer pGVm, Point 1, the gain is much larger on tasks with richer reasoning structure, such as math and coding, while on structurally simple tasks such as HQ the improvement over direct judging is small. This is exactly why we believe the contribution is not merely “graphizing” CoT, but using structure where structure actually matters.
>
> A second reason the problem is not trivial is that our data are already high-accuracy: for most settings, the underlying answer accuracy is above 90%. As a result, ROC-AUC is high for many methods, while AP remains much lower, because the task is to identify a relatively small number of hallucinated CoTs among mostly correct ones. In that regime, improving AP is not easy, and even modest absolute gains are meaningful.
>
> Moreover, as also reflected in your Point 2, the aggregation problem itself is not trivial. Different graph constructions and different aggregators behave quite differently in our ablations, and the same local verifier outputs do not lead to the same final ranking. This is why we introduce FLOW: not as a generic graph heuristic, but as a graph-coupled statistic whose value depends on how local evidence is propagated and combined under the inferred reasoning structure.
>
> Finally, when we inspected the ablations, we found that even a strong judge such as GPT-5.2 often misjudges because the evidence we provide is short, local, and semantically fragmented; in such cases, the judge may rely more on surface coherence than on the actual logic. This is consistent with prior work showing that LLM judges can exhibit verbosity bias, limited reasoning ability, and preference for superficial quality signals such as fluency or length (Zhou et al., _Mitigating the Bias of Large Language Model Evaluation_, CCL 2024). This makes our setting genuinely challenging, and further supports the need for a structured aggregation method rather than a single whole-trace judgment.

---

> > ### Author Rebuttal · Reviewer_U2nR · 2026-04-03
> >
> > I have read the rebuttal and decided to keep my score. The main reason is that the original paper only has two experiments. Plus the rebuttal, there are only three. I believe the paper is weak.

---

> > > ### Author Response · Authors · 2026-04-08
> > >
> > > Abbrev: G8=GSM8K, M=MATH, HQ=HotpotQA, HE=HumanEval; GPT=GPT-4o-mini, Gem=Gemini-2.5-Flash-Lite, Grok=Grok-4-Fast; FLOW+=Golden-DAG.
> > >
> > > We thank the reviewer for the suggestion. Below we provide additional experiments that separately examine structure, verifier strength, DAG quality, perturbation, difficulty, prompting, node sensitivity, and transfer, to clarify where SRM’s gains come from.
> > >
> > > **1) Judge-setting baseline.** We fix the verifier and change only evidence organization/aggregation.
> > >
> > > |Setting|Whole-lin|Whole-str|Prefix-step|FLOW|
> > > |---|---:|---:|---:|---:|
> > > |Base DAG|0.531|0.526|0.578|0.657|
> > > |Gold DAG|0.530|0.527|0.630|0.729|
> > >
> > > Here, Whole-lin/str=whole-trace judging with linear/structured input; Prefix-step=prefix-only local judging; Under the same verifier, FLOW is consistently above both whole-trace judging and prefix-only local judging, isolating the effect of structured evidence routing plus aggregation. The Base/Gold DAG contrast is also informative: better graph quality improves Prefix-step and FLOW together, indicating better support assignment for local judgment, while aggregation still adds gains beyond local scores alone.
> > >
> > > **2) Verifier ablation.** We replace the main verifier with a weaker local model.
> > >
> > > |Data|Var|Judge|GPT|Gem|Grok|Avg|
> > > |---|---|---|---:|---:|---:|---:|
> > > |G8|FLOW|Qwen|84.70|72.80|71.00|76.17|
> > > |G8|FLOW|GPT-5.2|66.20|74.80|66.23|69.08|
> > > |G8|FLOW+|Qwen|81.80|80.00|88.10|83.30|
> > > |G8|FLOW+|GPT-5.2|80.64|73.04|79.74|77.81|
> > > |M|FLOW|Qwen|62.40|67.00|59.60|63.00|
> > > |M|FLOW|GPT-5.2|63.46|73.17|66.14|67.59|
> > > |M|FLOW+|Qwen|78.40|68.90|62.90|70.07|
> > > |M|FLOW+|GPT-5.2|78.57|76.50|75.57|76.88|
> > >
> > > The method remains strong with Qwen-3.5-9B, showing that it is not tied to a single closed-source verifier. The ranking also varies across verifier and graph settings, suggesting that performance depends on the interaction between local judge quality and evidence structure. FLOW+ often improves substantially over FLOW, indicating that stronger graph structure can offset local judging noise.
> > >
> > > **3) DAG quality ablation.** We keep the pipeline fixed and vary only the graph source.
> > >
> > > |Data|Model|raw|qwen|gpt-5.2|
> > > |---|---|---:|---:|---:|
> > > |G8|GPT|84.70|81.41|81.80|
> > > |G8|Gem|72.80|68.21|80.00|
> > > |G8|Grok|71.00|80.70|88.10|
> > > |M|GPT|62.40|68.23|78.40|
> > > |M|Gem|67.00|65.05|68.90|
> > > |M|Grok|59.60|60.06|62.90|
> > >
> > > Better DAGs usually improve AP, sometimes substantially, indicating that graph quality directly affects final estimation.
> > >
> > > **4) Stability under DAG perturbation.** We perturb the evidence graph while keeping the FLOW aggregation rule fixed.
> > >
> > > |Perturbation|G8|M|
> > > |---|---:|---:|
> > > |Original|0.6424|0.5591|
> > > |Drop 10%|0.5126|0.4486|
> > > |Drop 20%|0.4067|0.3438|
> > > |Drop 50%|0.2317|0.2119|
> > > |Add 10%|0.6392|0.5438|
> > > |Add 20%|0.6368|0.5374|
> > > |Add 50%|0.6349|0.5296|
> > > |Drop cond.|0.0746|0.2112|
> > >
> > > Removing edges consistently lowers AP, and removing condition edges causes the largest drop. Adding random edges is less harmful, but still degrades AP, especially on M. This suggests that both missing and noisy support matter, with missing essential support being more damaging.
> > >
> > > **5) Task difficulty / CoT length.** We bucket examples by number of reasoning steps.
> > >
> > > |Steps|n|FLOW|Mean|Max|F-Max|
> > > |---|---:|---:|---:|---:|---:|
> > > |1-2|1319|0.628|0.604|0.607|0.021|
> > > |3|3471|0.565|0.416|0.517|0.048|
> > > |4|3711|0.582|0.414|0.494|0.088|
> > > |5|3048|0.560|0.372|0.471|0.089|
> > > |6|2189|0.574|0.345|0.437|0.137|
> > > |7|1321|0.599|0.392|0.486|0.113|
> > > |8+|2648|0.579|0.404|0.487|0.092|
> > >
> > > FLOW is only slightly above Max on short CoTs, but the gap widens on longer chains. This is consistent with aggregation becoming more important as local judgments become noisier over longer reasoning trajectories.
> > >
> > > **6) Structured prompting.** We compare generation accuracy with and without structured prompting.
> > >
> > > |Data|Plain|Struct.|
> > > |---|---:|---:|
> > > |G8 avg.|92.34|91.63|
> > > |HE avg.|86.59|90.65|
> > > |M avg.|74.87|74.00|
> > > |HQ avg.|74.89|67.25|
> > >
> > > Structured prompting is beneficial on difficult tasks such as HE, but is less effective on short-CoT tasks such as HQ, so the AP gains above cannot be explained by generation effects alone.
> > >
> > > **7) Node sensitivity.** We flip one node label in each importance bin rho and measure mean $\Delta$FLOW.
> > >
> > > |Data|rho1|rho2|rho3|rho4|
> > > |---|---:|---:|---:|---:|
> > > |G8|0.455|0.626|0.825|0.937|
> > > |M|0.186|0.275|0.444|0.686|
> > > |HQ|0.145|--|0.208|0.408|
> > > |HE|0.156|0.157|0.233|0.567|
> > >
> > > Higher-rho nodes have larger downstream influence, showing that structural position changes the impact of local errors.
> > >
> > > **8) Transfer beyond math.**
> > >
> > > |Data|Model|FLOW|Direct|
> > > |---|---|---:|---:|
> > > |HQ|Gem|0.848|0.823|
> > > |HQ|GPT|0.827|0.834|
> > > |HQ|Grok|0.801|0.767|
> > > |HE|Gem|0.819|0.306|
> > > |HE|GPT|0.777|0.574|
> > > |HE|Grok|0.698|0.528|
> > >
> > > The improvement is much larger on HE than on HQ, consistent with stronger benefits on tasks with longer and more structured reasoning.

---

### Official Review · Reviewer_pGVm · 2026-03-14

**Soundness:** 2
**Presentation:** 4
**Significance:** 3
**Originality:** 3
**Overall Recommendation:** 4
**Confidence:** 3

**Summary:**

This paper proposes a way to detect hallucinations by turning a chain-of-thought into a structured reasoning graph. The model writes short “conditions” and “steps,” and each step points to the earlier statements it relies on. The method then checks each step against its cited parents and combines those checks with a simple “flow” score over the graph.

**Compliance With Llm Reviewing Policy:**

Affirmed.

**Final Justification:**

The authors demonstrated they are open to my suggestions and ran many experiments to reduce my concerns.
I have updated my score accordingly.

**Key Questions For Authors:**

See my weaknesses

**Limitations:**

Yes

**Strengths And Weaknesses:**

Strengths:

* Moving from a linear CoT to an explicit dependency DAG is a sensible design choice. It makes “what supports what” explicit and enables step-level, local verification instead of judging the whole trace at once.

* The overall pipeline is straightforward and easy to implement.

* The results on the evaluated benchmarks look promising.

Weaknesses:

* The evaluation scope is quite limited: it focuses only on math reasoning (GSM8K/MATH), with a small number of datasets, and it’s unclear how well the approach transfers to other domains (coding, for example).

* The pipeline relies heavily on step-level LLM verification; performance degrades substantially with weaker judges, and open-source judge results drop sharply. This makes it hard to disentangle how much of the improvement comes from the proposed structure vs. judge capability.

* Reproducibility and future-proofing are concerns given the dependence on strong proprietary judges/backbones. Even though the setup is “black-box,” the authors could still include open-source backbones by using their outputs without accessing internal activations/attention, which would make the results easier to replicate and validate.

---

> ### Author Rebuttal · Authors · 2026-03-31
>
> Abbrev. G8=GSM8K, M=MATH, HQ=HotpotQA, HE=HumanEval. GPT=GPT-4o-mini, Gem=Gemini-2.5-Flash-Lite, Grok=Grok-4-Fast. FLOW+=Golden-DAG variant.
>
> **1. Transfer beyond math.**
> Thanks for the question. We completed HE/HQ transfer experiments and will add the results and analysis to the revision. The pattern is clear: our advantage over direct whole-trace judging is larger on tasks with more complex reasoning structure, such as math and coding, and smaller but still generally positive on shorter, structurally simpler tasks such as HQ. On HE, where solutions usually require longer multi-step program construction and more complex intermediate dependencies, FLOW consistently and substantially outperforms the direct-judge baseline across all three backbones. This matches our intuition that SRM helps most when errors arise at intermediate steps and must be localized and aggregated structurally, not judged only from the final trace. On HQ, reasoning chains are usually shorter and simpler, so the gap to direct judging is naturally smaller. Even so, FLOW still improves over direct judging for Gem and Grok, and remains competitive overall. We will add both the results and this complexity-based analysis to clarify where gains are largest and why.
>
> |Data|Model|FLOW|DirectJudge|
> |---|---|--:|--:|
> |HQ|Gem|0.848|0.823|
> |HQ|GPT|0.827|0.834|
> |HQ|Grok|0.801|0.767|
> |HE|Gem|0.819|0.306|
> |HE|GPT|0.777|0.574|
> |HE|Grok|0.698|0.528|
>
> _Table. Same SRM pipeline. Gains are much larger on structurally complex tasks (e.g., coding) than on shorter tasks such as HQ._
>
> **2. Reproducibility with an weaker open-source verifier.**
> Thanks for the question. We agree that judge dependence and reproducibility are central. To address this, we ran the same pipeline with a fully local open-source verifier, Qwen-3.5-9B, and compared it directly with the GPT-5.2 verifier used in the main paper on the two main benchmarks. We will add these results and discussion to the revision.
>
> |Data|Variant|Judge|GPT|Gem.|Grok|Avg.|
> |---|---|---|--:|--:|--:|--:|
> |GSM8K|FLOW|Qwen-3.5-9B|84.70|72.80|71.00|76.17|
> |GSM8K|FLOW|GPT-5.2|66.20|74.80|66.23|69.08|
> |GSM8K|FLOW+|Qwen-3.5-9B|81.80|80.00|88.10|83.30|
> |GSM8K|FLOW+|GPT-5.2|80.64|73.04|79.74|77.81|
> |MATH|FLOW|Qwen-3.5-9B|62.40|67.00|59.60|63.00|
> |MATH|FLOW|GPT-5.2|63.46|73.17|66.14|67.59|
> |MATH|FLOW+|Qwen-3.5-9B|78.40|68.90|62.90|70.07|
> |MATH|FLOW+|GPT-5.2|78.57|76.50|75.57|76.88|
>
> _Table. Main-paper benchmarks with two verifiers under the same SRM pipeline (AP; higher is better)._
>
> These results show that our method does not rely on closed-source APIs or GPT-5.2 specifically. With the same SRM pipeline, a fully local open-source verifier still achieves strong performance: on GSM8K, Qwen-3.5-9B is stronger on average than GPT-5.2 for both FLOW and FLOW+; on MATH, it is somewhat lower, but still clearly above the strongest non-FLOW baselines in the paper. More broadly, our method is black-box and interface-level: it uses only structured text plus local verification, and does not require hidden states or attention access.
>
> To isolate judge strength from structured decomposition/aggregation, we also add a same-verifier ablation on Grok/MATH, where the verifier is fixed and only the evidence organization/scoring protocol changes. Whole-trace scoring stays near 0.53 AP, whereas citation-grounded local step verification with FLOW aggregation improves substantially.
>
> |Setting|Whole-linear|Whole-struct|Prefix-step|Step-FLOW|
> |---|--:|--:|--:|--:|
> |Base DAG|0.531|0.526|0.578|0.657|
> |Gold DAG|0.530|0.527|0.630|0.729|
>
> _Table. Same-verifier ablation on Grok/MATH: identical judge, different evidence organization and aggregation._
>
> This same-judge result indicates that a substantial part of the gain comes from how evidence is decomposed, grounded, and aggregated, not merely from swapping in a stronger judge. We will add this ablation and the open-source-verifier results to make the attribution and reproducibility story clearer.
>
> **3. Reproducibility / open-source verifiers.**
> Thanks for this important point. We have already added results with a fully local open-source verifier (Qwen-3.5-9B), showing that our method does not rely on strong closed-source judging alone. In the revision, we plan to strengthen the reproducibility package by releasing the sampled reasoning traces in the supplement, allowing reviewers and future readers to inspect the structured inputs used by our pipeline and to reproduce the verification and aggregation stage more easily. Because our framework is purely black-box and interface-level, it naturally accommodates both open-source generators and open-source verifiers. We further plan to explore open-source models for reasoning-trace generation, which would make the full pipeline easier to replicate end-to-end without relying on proprietary systems.

---

> > ### Author Rebuttal · Reviewer_pGVm · 2026-04-02
> >
> > The authors demonstrated they are open to my suggestions and ran many experiments to reduce my concerns.
> > I have updated my score accordingly.

---

> > > ### Author Response · Authors · 2026-04-08
> > >
> > > Thank you very much for your thoughtful follow-up, for revisiting our rebuttal in detail, and for raising the score. We are grateful that you found our additional experiments and clarifications helpful.
> > >
> > > We especially appreciate your recognition that the paper’s main concerns were addressed through the added analyses on transfer beyond math, judge dependence, and reproducibility. Your suggestions helped us substantially strengthen the paper. In the revision, we will incorporate these new results and discussions directly into the main text, including:
> > >
> > > (1) the HumanEval/HotpotQA transfer results and the corresponding complexity-based analysis;
> > >
> > > (2) the open-source verifier results with Qwen-3.5-9B;
> > >
> > > (3) the same-verifier ablation that better isolates the contribution of structured decomposition and aggregation from judge strength.
> > >
> > > We will also further strengthen the reproducibility discussion and supplement materials as noted in our rebuttal.
> > >
> > > Thank you again for your time, careful reading, and constructive feedback. We sincerely appreciate your engagement with the paper.

---

### Decision · Program_Chairs · 2026-04-30

**Decision:**

Accept (regular)

**Comment:**

This paper proposes a hallucination detection method which converts a chain-of-thought into a structured reasoning graph.

- Reviewers agree that an explicit dependency DAG with step-level verification should be more reliable than a linear CoT for hallucination detection.

-The interpretable theoretical framework and good performance looks impressive.

- However, reviewers also agree that the evaluation scope of this paper is limited in only math reasoning only, though hallucination detection is a common issue for a wide range of tasks. The authors did provide preliminary results on other tasks during rebuttal which should be further extended in the revision.

- Another concern is about the usage of proprietary models for detection. Whether open-sourced LLMs can also achieve camparable performance is doubtable.